# APP β-CTF triggers cell-autonomous synaptic toxicity independent of Aβ

Mengxun Luo[1,2], Jia Zhou[1,2], Cailu Sun[1,2], Wanjia Chen[1,2], Chaoying Fu[1], Chenfang Si[1,2], Yaoyang Zhang[1], Yang Geng[1], Yelin Chen[1]*

[1]Interdisciplinary Research Center on Biology and Chemistry, Shanghai Institute of Organic Chemistry, Chinese Academy of Sciences, Shanghai, China; [2]University of Chinese Academy of Sciences, Beijing, China

## eLife Assessment

This study presents a **useful** demonstration that a specific protein fragment may induce the loss of synapses in Alzheimer's disease. The evidence supporting the data is **solid** but only partially supports the conclusion and would benefit from additional discussion indicated by the literature from reviewer #1. The application of the findings is limited because blocking the formation of the protein fragment has not benefited patients in several clinical trials.

*For correspondence:
chenyelin@sioc.ac.cn

**Abstract** Aβ is believed to play a significant role in synaptic degeneration observed in Alzheimer's disease and is primarily investigated as a secreted peptide. However, the contribution of intracellular Aβ or other cleavage products of its precursor protein (APP) to synaptic loss remains uncertain. In this study, we conducted a systematic examination of their cell-autonomous impact using a sparse expression system in rat hippocampal slice culture. Here, these proteins/peptides were overexpressed in a single neuron, surrounded by thousands of untransfected neurons. Surprisingly, we found that APP induced dendritic spine loss only when co-expressed with BACE1. This effect was mediated by β-CTF, a β-cleavage product of APP, through an endosome-related pathway independent of Aβ. Neuronal expression of β-CTF in mouse brains resulted in defective synaptic transmission and cognitive impairments, even in the absence of amyloid plaques. These findings unveil a β-CTF-initiated mechanism driving synaptic toxicity irrespective of amyloid plaque formation and suggest a potential intervention by inhibiting the endosomal GTPase Rab5.

## Introduction

Alzheimer's disease (AD) stands as the most prevalent form of neurodegenerative disease and a leading cause of dementia. It is characterized by extracellular amyloid plaque deposition, intracellular neurofibrillary tangles, synaptic and neuronal loss, and neuroinflammation (*Blennow et al., 2006*; *Gómez-Isla et al., 1996*; *Hardy and Selkoe, 2002*; *Leyns and Holtzman, 2017*; *Selkoe, 2002*). The amyloid plaque primarily consists of aggregated Aβ, a cleavage product of amyloid precursor protein (APP) (*Glenner and Wong, 1984*; *Masters et al., 1985*). Notably, two Aβ antibodies have demonstrated efficacy in removing amyloid plaques and slowing AD progression in phase III clinical trials (*Sims et al., 2023*; *van Dyck et al., 2023*). Furthermore, numerous naturally occurring mutations in genes encoding APP or its catalytic enzyme γ-secretase result in early-onset familial AD or reduce AD risk (*Jonsson et al., 2012*; *Liu et al., 2017*; *Mullan et al., 1992*; *Nilsberth et al., 2001*). This collective evidence, spanning AD pathology, human genetics, and intervention trials, strongly supports a causal role of Aβ and amyloid plaque in AD pathogenesis. However, despite clinical trials employing Aβ antibodies targeting Aβ oligomers, protofibrils, or deposited plaque, AD progression has been

slowed down by only ~30% (*Mintun et al., 2021*; *Pleen and Townley, 2022*; *Sims et al., 2023*; *van Dyck et al., 2023*). Notably, antibodies against monomeric soluble Aβ failed to yield clinical benefits (*Sperling et al., 2023*). It is possible that these Aβ antibodies may overlook certain pathogenic factors crucial for AD pathogenesis.

APP, a type I transmembrane protein, undergoes cleavage primarily by α-secretase on the cytoplasmic membrane, producing soluble α-cleavage N-terminal fragment (sAPPα) and α-cleavage C-terminal fragment (α-CTF). Some APP molecules bypass α-cleavage and undergo endocytosis into endocytic compartments, where they are subsequently cleaved by β-secretase, generating soluble β-cleavage N-terminal fragment (sAPPβ) and β-cleavage C-terminal fragment (β-CTF). β-CTF is further cleaved by γ-secretase to produce Aβ and APP intracellular domain (AICD) (*Golde et al., 1992*; *Zhang and Song, 2013*; *Zhang et al., 2011*). While extracellular Aβ, targeted by Aβ antibodies, is widely studied, the potential contribution of intracellular Aβ, APP, and other APP cleavage products to AD pathogenesis remains uncertain (*Konietzko, 2012*; *Kwart et al., 2019*; *Nikolaev et al., 2009*; *Oddo et al., 2003*; *Vohra et al., 2010*; *Willem et al., 2015*). For instance, β-CTF has been implicated in endosomal dysfunction (*Israel et al., 2012*; *Jiang et al., 2010*; *Kim et al., 2016*; *Kwart et al., 2019*; *Xu et al., 2016*), yet its downstream functional impacts remain unclear.

Synapse loss represents an early feature of AD neurodegeneration and is closely associated with cognitive dysfunction (*de Wilde et al., 2016*; *DeKosky et al., 1996*; *Terry et al., 1991*). Secreted Aβ induces synaptic dysfunction by interacting with its receptors on the neuronal plasma membrane (*Kamenetz et al., 2003*; *Kessels et al., 2013*; *Wei et al., 2010*). Other studies have reported γ-secretase inhibition reduced spine density in vivo via an APP-dependent pathway (*Bittner et al., 2009*). Additionally, deletion of APP in mice has been shown to decrease dendritic spine density (*Tyan et al., 2012*). The diverse outcomes of APP on synapse suggest a complex impact of APP and metabolites. It remains unclear whether APP or other APP fragments can also induce synaptic toxicity in a cell-autonomous manner.

To address these inquiries, we employed a sparse transfection system utilizing the Helios gene gun to explore the potential role of intracellular Aβ or other APP fragments. These molecules were expressed in a single neuron surrounded by untransfected wild-type neurons. Surprisingly, full-length APP did not induce synaptic toxicity. However, co-expression of APP with BACE1 resulted in significant loss of dendritic spines, indicating a crucial role of APP β-cleavage in synaptic damage. Further investigations unveiled that this detrimental effect was mediated by β-CTF in a cell-autonomous manner, independent of Aβ. Additionally, in vivo expression of β-CTF was adequate to induce synaptic dysfunction and cognitive impairments in mice, even in the absence of amyloid plaques. In summary, our study delineates a mechanism initiated by β-CTF that can induce synaptic degeneration in a cell-autonomous manner, thus extending beyond the scope of Aβ antibody-based therapies.

## Results

### APP only led to spine loss when co-expressed with BACE1

To investigate the cell-autonomous impact of APP on neurons, we utilized a Helios gene gun transfection system to sparsely express APP in rat organotypic hippocampal slice cultures (*Chen et al., 2014*). In hippocampal slice cultures at DIV9, transient expression of a familial AD APP mutation (APP$_{Swedish}$) (*Mullan et al., 1992*) with GFP in CA1 pyramidal neurons for 6 days did not reduce their dendritic spine densities compared with neurons expressing GFP alone (*Figure 1A and B*), suggesting that APP alone does not significantly contribute to synaptic loss.

BACE1-mediated β-cleavage of APP is crucial for amyloidogenesis, and BACE1 activity is notably elevated in AD patient brains (*Cheng et al., 2014*; *Vassar et al., 1999*). In human brains, APP and BACE1 are expressed at a ratio of about 15:1 (*Uhlén et al., 2015*). Endogenous BACE1 levels may not suffice to cleave the overexpressed APP. To assess whether insufficient β-cleavage of APP underlies the lack of synaptic toxicity caused by overexpressed APP, we co-expressed APP with BACE1 (in a 15:1 ratio) in organotypic hippocampal slice cultures and observed a significant ~40% reduction in spine density compared to neurons expressing GFP alone (*Figure 1C and D*). Transient expression of BACE1 alone did not affect spine density (*Figure 1E and F*). Co-expression of APP and BACE1 with different ratios or using an internal ribosome entry site (IRES) also resulted in significant spine loss

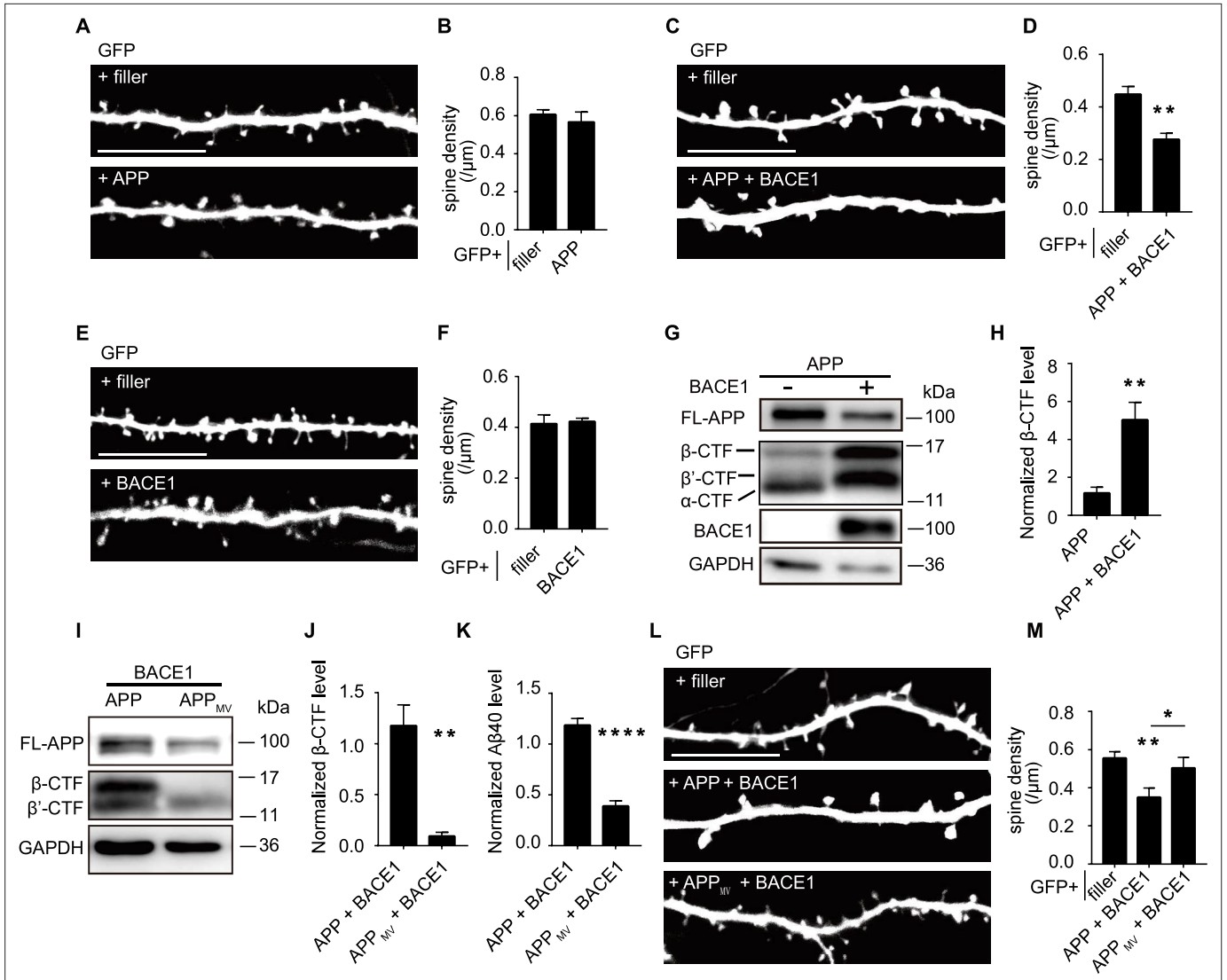

**Figure 1.** APP only led to spine loss when co-expressed with BACE1. (**A, B**) Representative images and spine density of basal dendrites from CA1 pyramidal neurons transiently expressing GFP alone or together with APP. filler, n=21; APP, n=16. (**C, D**) Representative images and spine density of basal dendrites from CA1 pyramidal neurons transiently expressing GFP alone or together with APP and BACE1 with a ratio of 15:1. filler, n=8; APP plus BACE1, n=5. (**E, F**) Representative images and spine density of basal dendrites from CA1 pyramidal neurons transiently expressing GFP alone or together with BACE1. filler, n=15; BACE1, n=15. (**G**) Western blot and corresponding statistical analysis of APP (Y188), BACE1, and GAPDH from HEK293T cells expressing APP or APP plus BACE1 with a ratio of 15:1. GAPDH was measured as a loading control. (**H**) Measurement of APP β-CTF (β C-terminal fragment) level from HEK293T cells expressing APP alone or APP and BACE1 with a ratio of 15:1. n=4. (**I**) Western blot and corresponding statistical analysis of APP fragments (Y188) from HEK293T cells expressing APP plus BACE1 or APP$_{MV}$ plus BACE1. (**J**) Measurement of APP β-CTF levels from HEK293T cells expressing APP plus BACE1 or APP$_{MV}$ plus BACE1. n=4. (**K**) Measurement of Aβ40 levels from HEK293T cells expressing APP plus BACE1 or APP$_{MV}$ plus BACE1. n=4. (**L, M**) Representative images and spine density of basal dendrites from CA1 pyramidal neurons transiently expressing GFP alone or together with APP and BACE1 or APP$_{MV}$ and BACE1. filler, n=12; APP plus BACE1, n=14; APP$_{MV}$ plus BACE1, n=10. All dendritic images were acquired from rat organotypic hippocampal slice cultures after transfection for 6–7 days. Statistics: one-way ANOVA or Student's *t*-test. *p<0.05, **p<0.01, ***p<0.001, ****p<0.0001. Error bars show SEM. Scale bars, 10 μm.

The online version of this article includes the following source data and figure supplement(s) for figure 1:

**Source data 1.** Original files for western blots shown in *Figure 1*.

**Source data 2.** Original files for western blots shown in *Figure 1*, indicating relevant bands.

**Source data 3.** Excel file containing numeric values for *Figure 1*.

**Figure supplement 1.** Different APP and BACE1 expression ratios impacted spine density differently.

**Figure supplement 1—source data 1.** Original files for western blots shown in *Figure 1—figure supplement 1*.

*Figure 1 continued on next page*

*Figure 1 continued*

**Figure supplement 1—source data 2.** Original files for western blots shown in *Figure 1—figure supplement 1*, indicating relevant bands.

**Figure supplement 1—source data 3.** Excel file containing numeric values for *Figure 1—figure supplement 1*.

(*Figure 1—figure supplement 1A and B*). These findings support the requirement of BACE1 for APP to induce synaptic loss.

When expressed alone in HEK293T cells, APP is predominantly cleaved at its α site to produce sAPPα and α-CTF (C-terminal fragment after APP α-cleavage) (*Figure 1G*, *Figure 1—figure supplement 1C*). Co-expression with BACE1 led to increased cleavage of APP at its β and β' sites, resulting in elevated β-CTF (more than fourfold) and β'-CTF as the major metabolic products and a substantial reduction in α-CTF levels (*Figure 1G and H*, *Figure 1—figure supplement 1C*).

APP$_{MV}$ (M596V) cannot be cleaved by BACE1 to produce β-CTF and Aβ but has no impact on β'-cleavage (*Figure 1I–K*; *Citron et al., 1995*). When co-expressed with BACE1, APP$_{MV}$ failed to induce spine loss (*Figure 1L and M*), supporting the requirement of β-cleavage of APP to induce spine loss.

In summary, these findings suggest that certain β-cleavage products of APP, rather than APP or BACE1 alone, could lead to spine loss in a cell-autonomous manner.

## β-CTF induced spine loss independent of Aβ

APP undergoes cleavage by secretases, generating soluble N-terminal fragments (sAPPs) and C-terminal fragment (CTFs). We then explored the impact of different sAPPs and APP-CTFs on dendritic spines. Transient expression of β-CTF significantly reduced the spine density of CA1 pyramidal neurons in organotypic rat hippocampal cultures, whereas expression of sAPPα, sAPPβ, α-CTF, or β'-CTF did not produce such an effect (*Figure 2A and B*, *Figure 2—figure supplement 1A and B*). APP CTFs exhibited similar expression levels in HEK293T cells (*Figure 2C*). When the plasmid amount was reduced to 1/8 of the original dose, β-CTF no longer induced a decrease in dendritic spine density (*Figure 2—figure supplement 1E and F*), indicating the synaptic damaging effect of β-CTF is its expression level dependent.

β-CTF undergoes processing by γ-secretase to generate Aβ and AICD (APP intracellular domain) (*Thinakaran and Koo, 2008*). Subsequently, we investigated whether Aβ generated from β-CTF was responsible for the β-CTF-induced spine loss. Treatment with a γ-secretase inhibitor, PF03084014 (1 μM), effectively reduced Aβ to baseline levels in cells expressing β-CTF without altering the expression levels of β-CTF itself (*Figure 2D and E*). PF03084014 treatment did not affect the spine density in neurons expressing GFP alone (*Figure 2—figure supplement 1C and D*). Notably, PF03084014 treatment failed to prevent the spine loss induced by β-CTF expression (*Figure 2F and G*) or by co-expression of APP and BACE1 (*Figure 2H–K*), suggesting that Aβ might not be the causative factor.

To delve deeper into the impact of Aβ on dendritic spines, we engineered two APP mutants (APP$_{\Delta59}$ and APP$_{\Delta57}$) lacking the AICD domain, which are known to generate significant amounts of Aβ40 and Aβ42, respectively (*Figure 2L and M*). Co-expression of either APP$_{\Delta59}$ or APP$_{\Delta57}$ with BACE1 did not alter spine density (*Figure 2N and O*), further bolstering the idea that Aβ is not responsible for the β-CTF-induced spine loss. Subsequently, we investigated the involvement of another β-CTF cleavage product, AICD, which has been reported to interact with the transcription factor forkhead box O (FoxO) and promote FoxO-induced transcription of proapoptotic genes, leading to cell death (*Wang et al., 2014*). However, transient expression of AICD failed to alter the density of spines (*Figure 2—figure supplement 1G and H*).

In conclusion, these results support the idea that APP can induce spine loss in a cell-autonomous manner through β-CTF, independent of Aβ and AICD.

## Expression of β-CTF damaged synapses in mice

To ascertain whether β-CTF-induced spine loss could manifest in vivo, we examined the density of dendritic spines from CA1 pyramidal neurons infected with lentivirus encoding β-CTF and GFP, or GFP alone, in adult mice. Sparse expression of GFP and β-CTF was observed in CA1 pyramidal neurons (*Figure 3A*). The spine density of CA1 pyramidal neurons expressing β-CTF and GFP was significantly lower than that of neurons expressing GFP alone (*Figure 3B and C*), indicating that β-CTF could induce spine loss in a cell-autonomous manner in vivo.

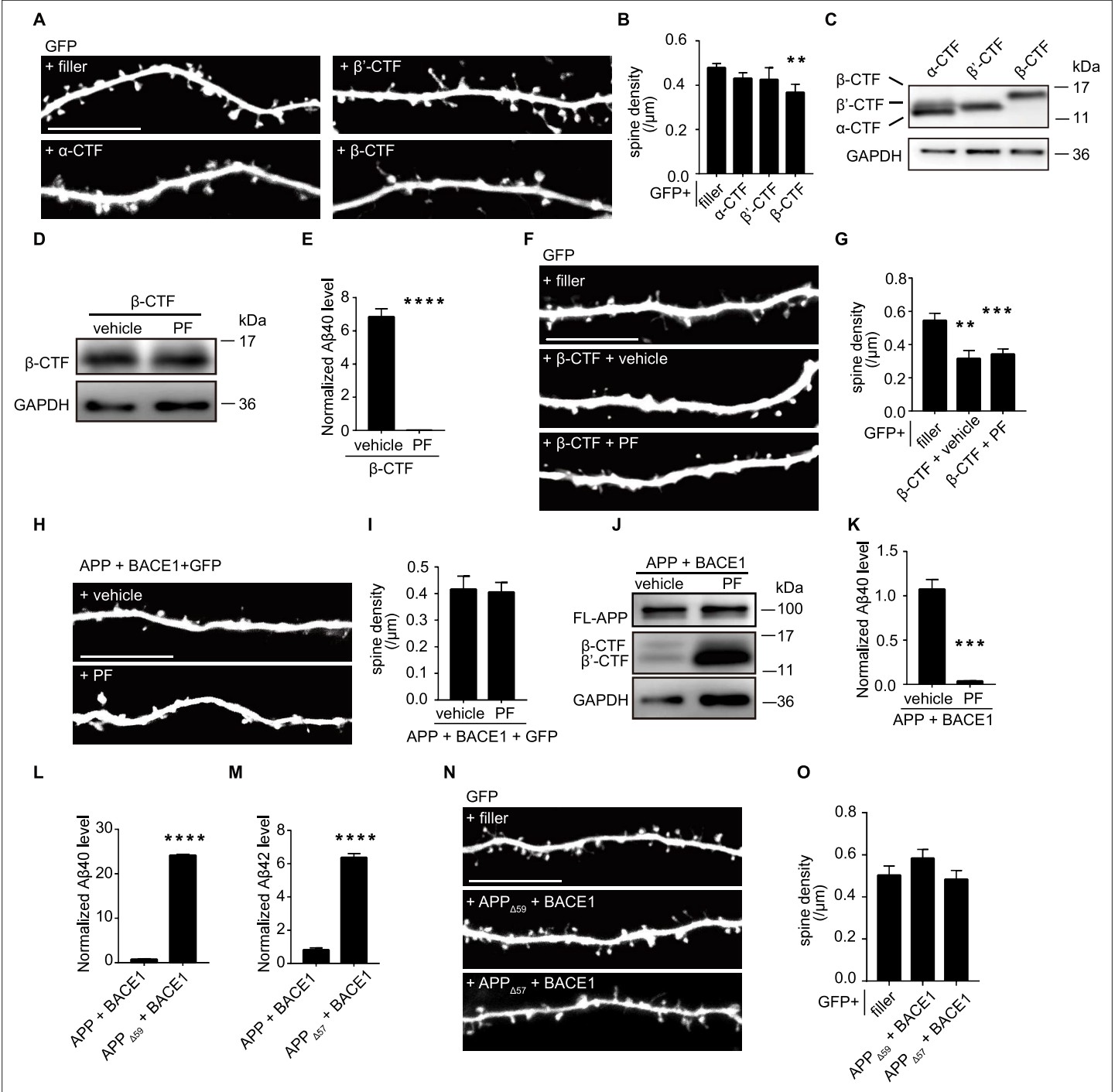

**Figure 2.** β-CTF induced spine loss independent of Aβ. (**A, B**) Representative images and spine density of basal dendrites from CA1 pyramidal neurons transiently expressing GFP alone or together with α-CTF, β'-CTF, or β-CTF. filler, n=19; α-CTF, n=7; β'-CTF, n=8; β-CTF, n=7. (**C**) Western blot of APP fragments from HEK293T cells expressing α-CTF-flag, β'-CTF-flag, and β-CTF-flag (flag antibody). (**D**) Western blot of APP fragments from HEK293T cells expressing β-CTF-flag treatment with vehicle or PF (flag antibody). (**E**) Measurement of Aβ40 secreted from HEK293T cells expressing β-CTF treatment with vehicle or PF. n=4. (**F, G**) Representative images and spine density of basal dendrites from CA1 pyramidal neurons transiently expressing GFP alone or together with β-CTF after treated with vehicle or PF. filler, n=11; β-CTF, n=7, β-CTF plus PF, n=14. (**H, I**) Representative images and spine density of basal dendrites from CA1 pyramidal neurons transiently expressing GFP together with APP and BACE1 after treated with vehicle or PF. APP plus BACE1 with vehicle, n=9; APP plus BACE1 with PF, n=14. (**J**) Western blot of APP fragments (Y188) from HEK293T cells co-expressing APP and BACE1 after treated with vehicle or PF. (**K**) Measurements of Aβ40 from HEK 293T cells expressing APP and BACE1 after treated with vehicle or PF. n=4. (**L**) Measurements of Aβ40 from HEK293T cells expressing APP and BACE1 or APP$_{\Delta59}$ and BACE1. n=4. (**M**) Measurements of Aβ42 from HEK293T cells

*Figure 2 continued on next page*

*Figure 2 continued*

expressing APP and BACE1 or APP$_{\Delta57}$ and BACE1. n=4. (**N, O**) Representative images and spine density of basal dendrites from CA1 pyramidal neurons transiently expressing GFP alone or together with APP$_{\Delta59/\Delta57}$ and BACE1. filler, n=9; APP$_{\Delta59}$ and BACE1, n=7; APP$_{\Delta57}$ and BACE1, n=6. All dendritic images were acquired from rat organotypic hippocampal slice cultures after transfection for 6–7 days. PF, PF03084014, a γ secretase inhibitor. Statistics: one-way ANOVA or Student's *t*-test. *p<0.05, **p<0.01, ***p<0.001, ****p<0.0001. Error bars show SEM. Scale bars, 10 μm.

The online version of this article includes the following source data and figure supplement(s) for figure 2:

**Source data 1.** Original files for western blots shown in *Figure 2*.

**Source data 2.** Original files for western blots shown in *Figure 2*, indicating relevant bands.

**Source data 3.** Excel file containing numeric values for *Figure 2*.

**Figure supplement 1.** sAPP or APP intracellular domain (AICD) expression and γ-secretase inhibition did not affect dendritic spines.

**Figure supplement 1—source data 1.** Excel file containing numeric values for *Figure 2—figure supplement 1*.

Next, we evaluated excitatory synaptic transmission through whole-cell patch-clamp recording of hippocampal pyramidal neurons infected with lentivirus encoding β-CTF or GFP. Neurons expressing β-CTF exhibited an ~65% lower frequency of miniature excitatory postsynaptic currents (mEPSCs) compared to neighboring uninfected neurons, whereas there was no significant change in mEPSC frequency in neurons expressing GFP alone (*Figure 3D and E*). However, mEPSC amplitude remained unaltered in neurons expressing either β-CTF or GFP (*Figure 3D and F*). These findings collectively suggest that β-CTF leads to reduced excitatory synapse density without affecting the strength of the remaining synapses.

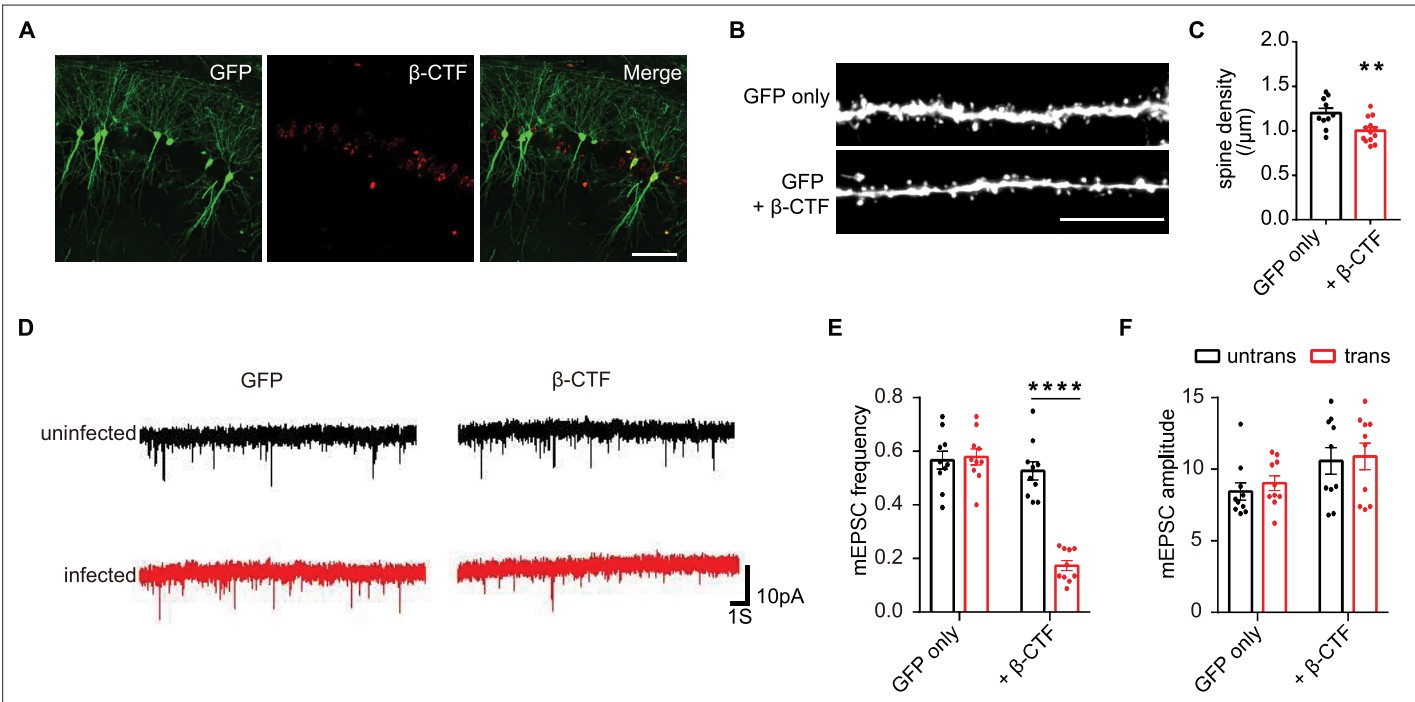

**Figure 3.** Expression of β-CTF damaged synapses in mouse brains. (**A**) Representative images of adult mouse CA1 pyramidal neurons infected with lentivirus expressing GFP and β-CTF. Scale bar, 100 μm. (**B, C**) Representative images from basal dendrites and quantitation of their spine density in CA1 pyramidal neurons infected with lentivirus expressing GFP alone or GFP with β-CTF in mouse hippocampi. Scale bar, 10 μm. GFP only, n=10; GFP plus β-CTF, n=13. (**D–F**) Representative recording traces and quantitations of miniature excitatory postsynaptic currents (mEPSCs) from adult mouse hippocampal neurons infected with lentivirus (identified with GFP signal) and neighboring uninfected neurons. n=10. Statistics: two-way ANOVA or Student's *t*-test. *p<0.05, **p<0.01, ***p<0.001, ****p<0.0001. Error bars show SEM.

The online version of this article includes the following source data for figure 3:

**Source data 1.** Excel file containing numeric values for *Figure 3*.

## Expression of β-CTF damaged cognitive function in mice in the absence of plaque formation

We proceeded to investigate whether β-CTF affects cognitive functions. Adeno-associated viruses (AAV) encoding GFP or β-CTF were bilaterally injected into the hippocampus of 1-month-old mice (*Figure 4A and B*). After 4 months of expression, animal behaviors were examined. Immunostaining revealed widespread expression of GFP or β-CTF throughout the entire hippocampus (*Figure 4B*), with no detectable amyloid plaque formation observed using ThS staining (*Figure 4C*). Side-by-side staining demonstrated robust amyloid plaque deposition in the brain of two-month-old 5XFAD mice (*Figure 4D*), validating the efficacy of the staining method. The body weight of mice expressing β-CTF in the hippocampus was approximately 17% lower than that of GFP controls (*Figure 4E*). In the Y maze test, the mean alternations were significantly reduced in mice expressing β-CTF compared to GFP controls, suggesting abnormal working memory in mice expressing β-CTF (*Figure 4F*).

The fear conditioning (FC) test assesses associative fear learning and memory (*Figure 4G*; *Xiao et al., 2018*). In this test, mice expressing β-CTF exhibited similar baseline levels of freezing time as GFP controls (*Figure 4H*). However, in the hippocampus-dependent contextual FC test, mice expressing β-CTF showed a significantly shorter freezing time (approximately 40% less) than those expressing GFP after training (*Figure 4H*). In the amygdala-dependent cued FC test, these two groups performed similarly (*Figure 4H*). In the water T maze test (*Figure 4I*), mice expressing β-CTF displayed slower learning curves than GFP controls, indicating impairments in acquisition learning (*Figure 4J*) but not in reversal learning (*Figure 4K*).

In the open-field test, mice expressing β-CTF traveled a longer distance (*Figure 4—figure supplement 1A–C*) and spent more time in the center area with a higher frequency of center entrances (*Figure 4—figure supplement 1D and E*) compared to mice expressing GFP. However, there were no differences in rearing frequency and duration between the two groups (*Figure 4—figure supplement 1F and G*). These results suggest that mice expressing β-CTF exhibited increased locomotion and reduced anxiety-like behaviors. Consistently, mice injected with AAV encoding β-CTF spent significantly more time exploring and traveled a greater distance in the open arms compared to the GFP controls in the elevated plus maze (EPM) (*Figure 4—figure supplement 1H–J*). In tail suspension tests (TSTs), these two groups exhibited similar levels of immobility (*Figure 4—figure supplement 1K*), indicating that β-CTF expression in the hippocampus did not alter depression-like behaviors.

In conclusion, these findings support that β-CTF expression is sufficient to disrupt hippocampus-dependent cognitive functions.

## The C-terminal YENPTY motif was necessary for β-CTF to induce endosomal dysfunction and synapse loss

Endosome abnormalities mediated by APP β-CTFs have been reported across various cell types, including human iPSC-induced neurons, PC12M cells, and N2a cells (*Kim et al., 2016*; *Kwart et al., 2019*; *Xu et al., 2016*). Next, we investigated whether β-CTF impacted endosomes in hippocampal neurons. Dissociated cultured hippocampal neurons were co-transfected with APP-CTFs and Rab5-GFP, an endosomal marker fused with GFP, at DIV7. In neurons expressing either Rab5-GFP alone or Rab5-GFP with α-CTF, Rab5 puncta appeared uniform and smoothly rounded (*Figure 5A and B*). However, in neurons co-transfected with β-CTF, Rab5 puncta were larger and exhibited less uniform shapes, often appearing lobular, and showed robust co-localization with β-CTF (*Figure 5C and D*). Notably, the morphology of lysosomes, as observed by Lamp1 staining, remained unaffected by the expression of α- or β-CTF in neurons (*Figure 5—figure supplement 1A–D*), thereby suggesting a specific interaction of β-CTF with endosomes.

The C-terminal YENPTY motif of APP was found to be crucial for its interaction with endosomes (*Lai et al., 1995*). To elucidate the significance of this interaction, we expressed a mutant form of β-CTF (β-CTF$_{mut}$), where the YENPTY motif was substituted with AENATA. Interestingly, β-CTF$_{mut}$ was expressed at similar levels to wildtype β-CTF and exhibited comparable Aβ production (*Figure 5E–H*). However, unlike wildtype β-CTF, β-CTF$_{mut}$ showed a more diffuse distribution throughout the neurons and failed to induce enlarged Rab5 puncta (*Figure 5I and J*). Notably, pyramidal neurons transiently expressing β-CTF$_{mut}$ displayed a higher spine density compared to those expressing wildtype β-CTF (*Figure 5K and L*). These findings underscore the critical role of the YENPTY motif-mediated interaction with endosomes in β-CTF-induced spine loss.

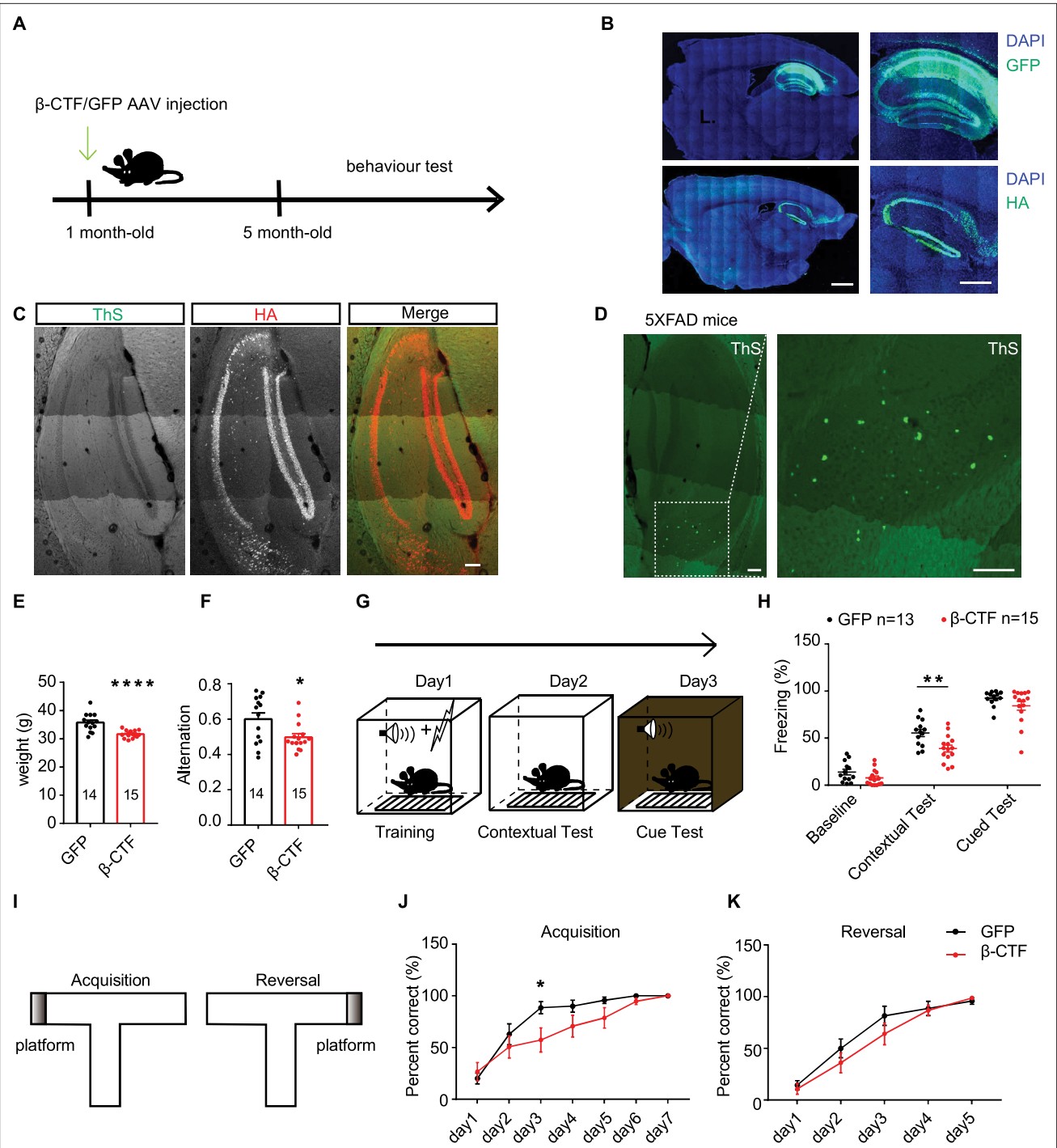

**Figure 4.** Expression of β-CTF damaged cognitive function in mice in the absence of plaque formation. (**A**) Schematic diagram showing the time line for stereotactic injection of adeno-associated viruses (AAV), behavioral training, and tests. (**B**) Representative images showing immunofluorescence staining of expressed GFP or β-CTF-HA by AAV in mouse hippocampi (WT). Scare bars represent 1000 μm (left) and 500 μm (right). (**C**) Images of mouse hippocampus (WT) stained with ThS and HA antibody after infected with AAV encoding β-CTF-HA. Scale bar, 100 μm. (**D**) Representative image of ThS staining of a 5XFAD mice hippocampal slice. Scale bar, 100 μm. (**E**) Weight of mice infected with AAV encoding GFP or β-CTF in their hippocampi. GFP, n=14; β-CTF, n=15. (**F**) Quantitations of spontaneous alternation in Y maze from mice infected with AAV encoding GFP or β-CTF. GFP, n=14; β-CTF, n=15. (**G**) Schematic diagram showing fear conditioning test design. (**H**) Quantitation of freezing in contextual and cued fear conditioning tests from mice infected with AAV encoding GFP and β-CTF in their hippocampi. GFP, n=13; β-CTF, n=15. (**I**) Schematic diagram showing water T maze test design. (**J, K**) The percentage of correct responses across the five trials of water T maze test was quantified in each day of acquisition (**J**) or reversal (**K**). GFP,

*Figure 4 continued on next page*

*Figure 4 continued*

n=14; β-CTF, n=15. Statistics: repeated measures two-way ANOVA or Student's *t*-test. *p<0.05, **p<0.01, ***p<0.001, ****p<0.0001. Error bars show SEM.

The online version of this article includes the following source data and figure supplement(s) for figure 4:

**Source data 1.** Excel file containing numeric values for *Figure 4*.

**Figure supplement 1.** Additional neurobehavioral tests of mice expressing β-CTF in their brains.

**Figure supplement 1—source data 1.** Excel file containing numeric values for *Figure 4—figure supplement 1*.

## Spine loss induced by β-CTF was prevented by Rab5 inhibition

To investigate the downstream mechanism responsible for β-CTF-induced synaptic loss, we analyzed the proteomic alterations triggered by β-CTF in dissociated cultured hippocampal neurons (*Figure 6A*). Our findings revealed significant changes in protein levels upon exposure to β-CTF (*Figure 6B*). Notably, the expression of Synapsin-1, a presynaptic protein associated with synaptic vesicles, and GluR1, GluN2A, GluN2B, subunits of glutamate receptors, were diminished in neurons expressing β-CTF (*Figure 6C–F*). VAMP2, a key component of SNARE complex, was also reduced by β-CTF (*Figure 6E and G*). The protein level of Synapsin-1 was further reduced after treatment with γ-secretase inhibitors in neurons expressing β-CTF (*Figure 6—figure supplement 1A and B*). Gene Ontology analysis further elucidated that β-CTF expression downregulated proteins involved in membrane trafficking, synaptic vesicle cycle, pre-synapse, and post-synapse (*Figure 6H and I*), aligning with previous observations indicating that β-CTF induces synaptic dysfunction and endosomal abnormalities (*Zhou et al., 2019*). Conversely, upregulated proteins were primarily associated with peptide metabolic processes and translation (*Figure 6—figure supplement 1C and D*).

Overexpression of β-CTF resulted in the enlargement of Rab5-positive endosomes, similar to the effects observed with a constitutively active mutant of Rab5, Rab5$_{Q79L}$ (*Figure 6—figure supplement 1E and F*; *Kim et al., 2016*). Notably, the introduction of a dominant negative mutant of Rab5, Rab5$_{S34N}$, attenuated the endosome enlargement induced by β-CTF (*Figure 6J and K*). In neurons expressing Rab5$_{S34N}$, there was reduced co-localization of β-CTF with Rab5 (*Figure 6J and K*). Critically, co-expression of Rab5$_{S34N}$ with β-CTF effectively mitigated the spine loss induced by β-CTF in hippocampal slice cultures (*Figure 6L and M*). These findings underscored that Rab5 overactivation-induced endosomal dysfunction contributed to β-CTF-induced spine loss. However, expression of Rab5$_{S34N}$ in β-CTF-expressing neurons did not alter the levels of synapse-related proteins that were reduced in these neurons (*Figure 6—figure supplement 1G and H*), suggesting Rab5 overactivation did not contribute to these protein expression changes induced by β-CTF.

## Discussion

Treatment with lecanemab or donanemab, two Aβ antibodies, significantly slowed down AD progression by approximately 30% in phase III clinical trials (*Sims et al., 2023*; *van Dyck et al., 2023*). The success of these antibodies in modifying AD progression validates Aβ as a cause of AD pathogenesis. However, the relatively modest benefits have raised additional questions. Despite effectively reducing amyloid plaques to near baseline levels after 18 months of treatment in some patients, functional benefits were limited to only ~30%. This raises the question of whether amyloid plaque is the primary source of Aβ-related toxicity or if these patients were treated too late to reverse their disease progression more effectively. This study aims to investigate whether Aβ-associated pathways could lead to pathogenesis beyond secreted Aβ and amyloid plaques, which falls outside the scope of Aβ antibody-based therapies due to their inability to penetrate cytoplasmic membranes.

AD is characterized by significant synaptic degeneration. Conventionally, Aβ is considered detrimental to synapses, thus synaptic loss in AD is attributed to various forms of Aβ, including amyloid plaques (*Kamenetz et al., 2003*; *Oakley et al., 2006*). However, conflicting studies have suggested that Aβ may actually promote synaptogenesis (*Abramov et al., 2009*; *Bittner et al., 2009*; *Puzzo et al., 2008*; *Zhou et al., 2022*). The exact role of Aβ in AD-related synaptic degeneration remains elusive. Most investigations have focused on the non-cell-autonomous function of Aβ after its secretion from neurons (*Reinders et al., 2016*). The potential contribution of intracellular Aβ or its precursors to synaptic toxicity has been largely unexplored. Utilizing a sparse neuron transfection system,

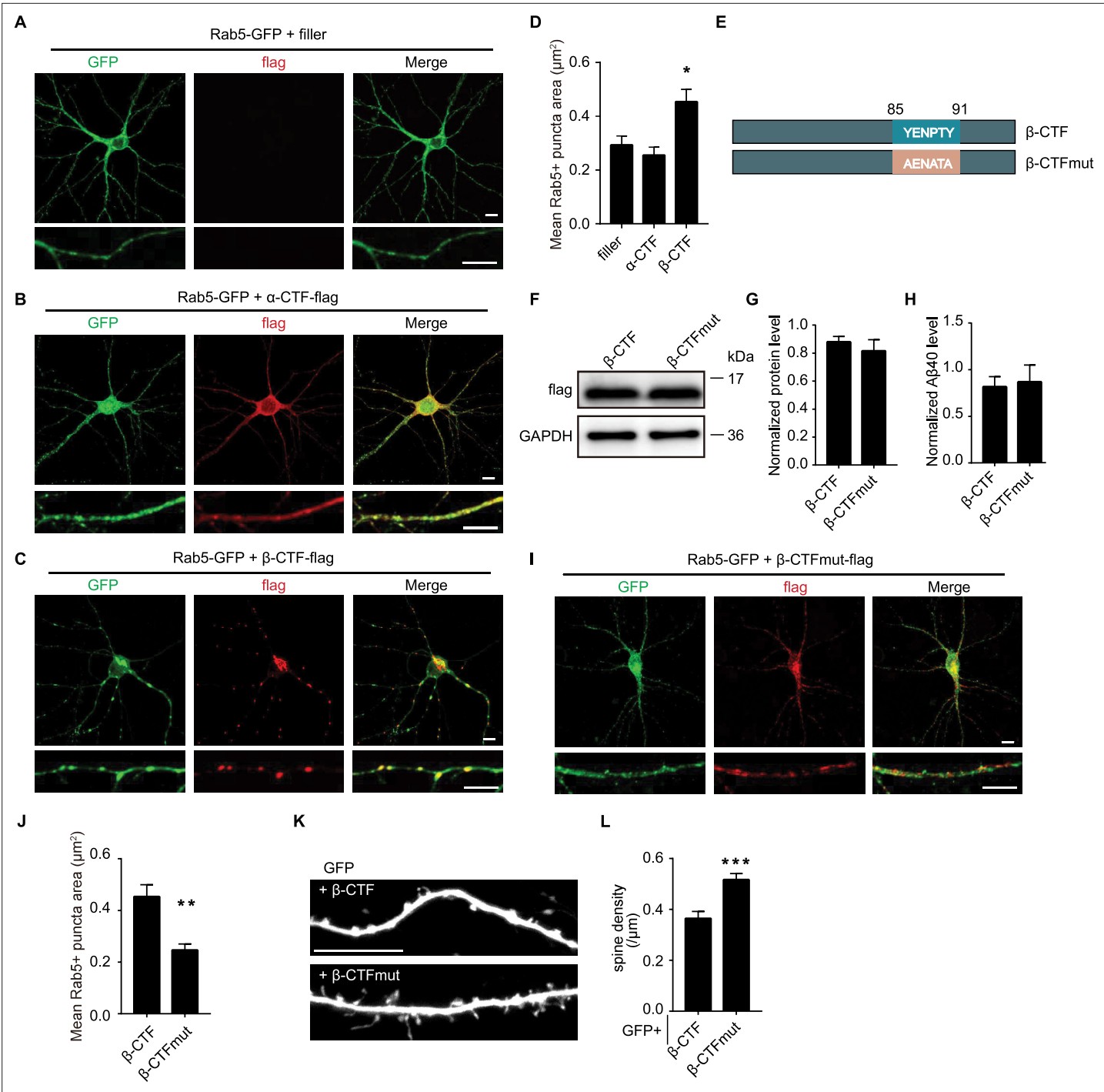

**Figure 5.** The C-terminal YENPTY motif was necessary for β-CTF to induce endosomal dysfunction and synapse loss. (**A**) Immunofluorescent staining of GFP (green) from dissociated rat hippocampal neurons expressing Rab5-GFP. (**B**) Immunofluorescent staining of GFP (green) and flag (red) from dissociated rat hippocampal neurons expressing Rab5-GFP and α-CTF-flag. (**C**) Immunofluorescent staining of GFP (green) and flag (red) from dissociated rat hippocampal neurons expressing Rab5-GFP and β-CTF-flag. (**D**) Quantitation of average Rab5+puncta size of neurons expressing Rab5-GFP only or co-expressing α/β-CTF-flag and Rab5-GFP. filler, n=9; α-CTF, n=9; β-CTF, n=11. (**E**) Schematic diagram of the β-CTF and β-CTFmut. (**F**) Western blot of APP fragments from HEK293T cells expressing β-CTF-flag and β-CTFmut-flag with a flag antibody. (**G**) Quantitations of APP fragments from HEK293T cells expressing β-CTF-flag and β-CTFmut-flag. n=4. (**H**) Measurement of Aβ40 from HEK 293T cells expressing β-CTF or β-CTFmut. n=5. (**I**) Immunofluorescent staining of GFP (green) and flag (red) from dissociated rat hippocampal neurons expressing Rab5-GFP and β-CTFmut-flag. (**J**) Quantitation of average Rab5+puncta size of neurons co-expressing β-CTF-flag or β-CTFmut-flag and Rab5-GFP. β-CTF, n=11; β-CTFmut, n=9. (**K, L**) Representative images and spine density of basal dendrites from CA1 pyramidal neurons transiently expressing GFP together

*Figure 5 continued on next page*

*Figure 5 continued*

with β-CTF or β-CTFmut for 6–7 days in rat organotypic hippocampal slice cultures. β-CTF, n=12; β-CTFmut, n=15. Scale bar, 10 µm. Statistics: one-way ANOVA or Student's *t*-test. *p<0.05, **p<0.01, ***p<0.001, ****p<0.0001. Error bars show SEM.

The online version of this article includes the following source data and figure supplement(s) for figure 5:

**Source data 1.** Original files for western blots shown in *Figure 5*.

**Source data 2.** Original files for western blots shown in *Figure 5*, indicating relevant bands.

**Source data 3.** Excel file containing numeric values for *Figure 5*.

**Figure supplement 1.** Co-immunostaining of APP CTFs and a lysosomal marker.

**Figure supplement 1—source data 1.** Excel file containing numeric values for *Figure 5—figure supplement 1*.

we systematically examined this question. Intriguingly, among Aβ, APP, and major APP cleavage products, only β-CTF induced synaptic toxicity in a cell-autonomous manner (*Figures 1 and 2*, *Figure 1—figure supplement 1*, *Figure 2—figure supplement 1*), aligning with predictions based on indirect evidence (*Kwart et al., 2019*; *Lee et al., 2022*; *Tamayev et al., 2012*; *Vaillant-Beuchot et al., 2021*; *Xu et al., 2016*). In mouse brains, β-CTF also triggered substantial synaptic loss and cognitive deficits in the absence of amyloid plaques (*Figure 4C*), further supporting the notion that synaptic loss and amyloid plaque formation are mediated by distinct mechanisms. While current data did not exclude the potential involvement of Aβ-induced toxicity in the synaptic and cognitive dysfunction observed in mice overexpressing β-CTF, addressing this directly remains challenging. Treatment with γ-secretase inhibitors could potentially shed light on this issue. However, treatments with γ-secretase inhibitors are known to lead to brain dysfunction by itself likely due to its blockade of the γ-cleavage of other essential molecules, such as Notch (*Doody et al., 2013*; *Güner and Lichtenthaler, 2020*), preventing from pursuing it further experimentally in vivo.

APP, Aβ, and presenilins have been extensively studied in mouse models, providing convincing evidence that high Aβ concentrations are toxic to synapses (*Chapman et al., 1999*). Moreover, addition of Aβ to murine cultured neurons or brain slices is toxic to synapses (*Wang et al., 2017*). However, Aβ-induced synaptotoxicity was not observed in our study. A major difference between our study and others is that we employed an isolated expression system, where Aβ was applied solely to individual neurons surrounded by other neurons, without overwhelming them with excessive amounts of Aβ. In contrast, other studies typically apply Aβ to neurons indiscriminately. Therefore, we predict that Aβ does not lead to synaptic deficits from individual neurons in cell-autonomous manners, whereas β-CTF does.

It is noteworthy that β-CTF induces spine loss independent of Aβ, implying that while synaptic degeneration and amyloidogenesis both occur downstream of β-CTF, they may represent two parallel processes independent of each other following APP β-cleavage. In fact, Aβ and β-CTF levels were significantly elevated in the AD brains (*Kim et al., 2016*). Aβ antibodies effectively facilitate the clearance of secreted Aβ, including that deposited in amyloid plaques. However, β-CTF localizes within the intracellular membrane of endo/lysosomal vesicles, rendering it inaccessible to Aβ antibodies. Consequently, Aβ antibody-based therapies cannot mitigate the neuronal toxicities initiated by β-CTF. This limitation could partially explain the restricted clinical benefits observed with Aβ antibodies. Addressing β-CTF-mediated synaptic toxicity should be prioritized in the future development of improved AD therapies, potentially in conjunction with Aβ antibodies.

However, clinical trials of BACE1 inhibitors have failed to demonstrate an efficacy in treating AD, despite their theoretical potential to inhibit the production of both β-CTF and Aβ. The underlying reason remains unclear. One possibility is that reduced BACE1 activity may diminish certain neurotrophic APP metabolites, such as AICD (*Ma et al., 2007*). Another potential reason is that pharmacological inhibition of BACE1 appears not as effective as its genetic removal. Genetic depletion of BACE1 leads to clearance of existing amyloid plaques (*Hu et al., 2018*), whereas pharmacological inhibition of BACE1 could not stop growth of existing plaques, although it slows down the growth of these plaques and prevents formation of new plaques (*Peters et al., 2018*). The development of better β-secretase inhibitors to more effectively inhibit β-cleavage is required to fully test the role of APP β-cleavage in AD pathogenesis.

Endosomal dysfunction is an early neuropathological signature of AD and Down syndrome (DS) mediated by β-CTF. Kim et al. found β-CTF recruited APPL1 (a Rab5 effector) via YENPTY motif to

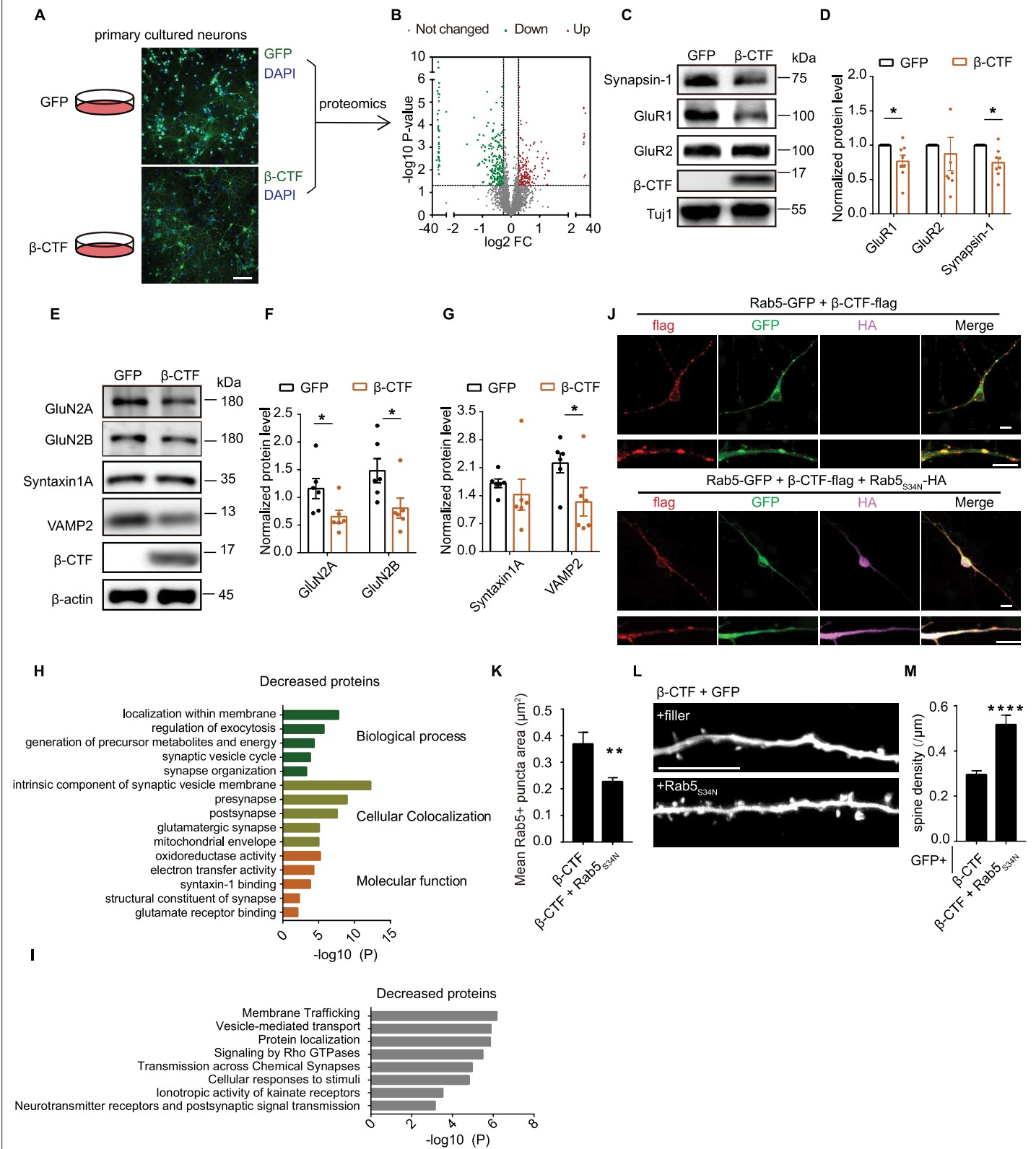

**Figure 6.** Spine loss induced by β-CTF was prevented by Rab5 inhibition. (**A**) Representative images of dissociated rat hippocampal neurons infected with lentivirus encoding GFP or β-CTF. Scale bar, 100 μm. (**B**) Volcano plot of quantitative mass spectrometry analysis showing protein omics changes in neurons infected with lentivirus encoding β-CTF v.s. GFP. Proteins with fold change >1.2 or <0.83 and p-value<0.05 are colored. (**C, D**) Western blot and corresponding statistic analysis of Synapsin-1, GluR1, GluR2 in dissociated hippocampal neurons infected with lentivirus expressing GFP or

*Figure 6 continued on next page*

*Figure 6 continued*

β-CTF. Tuj 1 was used as an internal control. n=8. (**E–G**) Western blot and corresponding statistic analysis of GluN2A, GluN2B, Syntaxin1A, VAMP2 in dissociated hippocampal neurons infected with lentivirus expressing GFP or β-CTF. β-actin was used as an internal control. n=6. (**H**) The Gene Ontology (GO) processes analysis of decreased proteins by Metascape. (**I**) The Reactome Gene Sets processes analysis of decreased protein by Metascape. (**J**) Immunofluorescent staining of GFP (green), flag (red) and HA (magenta) from dissociated rat hippocampal neurons expressing Rab5-GFP and β-CTF-flag or together with Rab5$_{S34N}$-HA. Scale bar, 10 μm. (**K**) Quantitation of average Rab5+puncta size from dissociated rat hippocampal neurons expressing Rab5-GFP and β-CTF-flag or together with Rab5$_{S34N}$-HA. β-CTF, n=5; β-CTF + Rab5$_{S34N}$, n=9. (**L, M**) Representative images and measurements of spine density of basal dendrites from CA1 pyramidal neurons transiently expressing GFP and β-CTF with or without Rab5$_{S34N}$ for 6–7 days in rat organotypic hippocampal slice cultures. Scale bar, 10 μm. β-CTF, n=17; β-CTF plus Rab5$_{S34N}$, n=11. Statistics: Student's *t*-test. *p<0.05, **p<0.01, ***p<0.001, ****p<0.0001. Error bars show SEM.

The online version of this article includes the following source data and figure supplement(s) for figure 6:

**Source data 1.** Original files for western blots shown in *Figure 6*.

**Source data 2.** Original files for western blots shown in *Figure 6*, indicating relevant bands.

**Source data 3.** Excel file containing numeric values for *Figure 6*.

**Figure supplement 1.** GO and Reactome Gene sets analysis of β-CTF-upregulated proteins in neurons.

**Figure supplement 1—source data 1.** Original files for western blots shown in *Figure 6—figure supplement 1*.

**Figure supplement 1—source data 2.** Original files for western blots shown in *Figure 6—figure supplement 1*, indicating relevant bands.

**Figure supplement 1—source data 3.** Excel file containing numeric values for *Figure 6—figure supplement 1*.

Rab5 endosomes, where it stabilizes active GTP-Rab5, leading to pathologically accelerated endocytosis, endosome swelling, and selectively impaired transport of Rab5 endosomes (*Kim et al., 2016*). Rab5 endosomal dysfunction may trigger prodromal and neurodegenerative features of AD (*Pensalfini et al., 2020*). However, the downstream functional impacts of endosomal dysfunction induced by β-CTF remain unclear. One of the impacts is deficits of NGF signaling in basal forebrain cholinergic neurons (BFCNs) (*Salehi et al., 2006*; *Xu et al., 2016*). Here, we report endosomal dysfunction-induced by β-CTF also contributes to synaptic dysfunction. Synaptic abnormalities are corrected by inhibiting Rab5 activity to restore endosomal function (*Figure 6J–M*), indicating that endosomal abnormalities may serve as a potential target for AD treatment.

β-CTF elicited significant spine loss, whereas α-CTF or β'-CTF, which are only 16 or 10 amino acids shorter at their N-terminal ends compared to β-CTF, did not exhibit this detrimental effect. Since the C-terminal YENPTY motif of β-CTF is essential for this activity, likely through its interaction with endosomal proteins, β-CTF represents the minimum size of the APP fragment capable of inducing cell-autonomous synaptic toxicity. However, the specific contribution of the N-terminal domain of β-CTF to synaptic loss remains unclear. It remains to be investigated whether longer fragments containing β-CTF could also induce synaptic loss. If confirmed, strategies aimed at reducing APP β-cleavage while promoting α- or β'-cleavage to decrease both Aβ and β-CTF may offer a more effective treatment for AD in the future.

## Materials and methods
### Dissociated hippocampal neuron culture and transfection

We prepared hippocampal neurons from embryonic day 17/18 (E17/18) SD rats. Initially, whole hippocampi were dissected under stereomicroscopes and then digested into single cells using papain (Worthington) for 25 minutes at 37°C. The dissociated cells were seeded at a density of 180,000 cells per well in a 24-well plate, using Neurobasal media (Thermo Fisher Scientific) supplemented with 2% B27 (Life technology), 0.25% GlutaMAX (Thermo Fisher Scientific), and 100 units/mL Penicillin/Streptomycin (Life Technology). Neurons were cultured in incubators maintained at 37°C with 5% $CO_2$, with fresh media half replenished every 5 days. For virus infection, dissociated hippocampal neurons at day in vitro (DIV) 14 were exposed to purified lentivirus. After 5 days, the infected neurons were lysed in RIPA buffer (Solarbio) supplemented with a protease inhibitor cocktail (Selleck) for further analysis. For plasmid transfection, dissociated hippocampal neurons at DIV7 were transfected with Lipofectamine 2000 (Thermo Fisher Scientific) following the manufacturer's instructions.

## Western blotting and immunofluorescent staining

For western blotting, we first removed the culture medium, and then immediately placed neurons or HEK293T cells on ice. Subsequently, they were lysed using 2×loading buffer (100 mM Tris–HCl, 150 mM NaCl, 4% SDS, 20% glycerol). The lysates were agitated for 10 minutes at room temperature, followed by boiling at 98°C for 10 minutes and centrifugation at 14,000 × $g$ for 10 minutes. The proteins were separated using 8–12% Tris-glycine SDS-PAGE and transferred to a PVDF membrane (Millipore, 0.22 μm). Primary antibodies were then incubated in blocking buffer overnight at 4°C, followed by secondary antibodies at room temperature for 2 hours. The signal was detected using an Amersham Imager 600 (GE Healthcare), and densitometry was measured using ImageJ.

For immunofluorescent staining, we first washed neurons on cover glasses with PBS (supplemented with 1 mM $CaCl_2$ and 5 mM $MgCl_2$) once. Then, we permeabilized them with 0.15% Triton-X in PBS for 15 minutes. After blocked with 5% bovine serum albumin in PBS for 30 minutes at room temperature, the samples were incubated with primary antibodies in blocking buffer at 4°C overnight. The following day, the samples were washed three times with PBS and then incubated with secondary antibodies at room temperature for 1 hour in the dark. Subsequently, the samples were washed three times with PBS and mounted in Mounting Medium with DAPI (Solarbio). Finally, images were acquired using an Andor spinning disk or Nikon confocal microscopy system. All images were captured and analyzed in a blinded manner.

## Organotypic hippocampal slice culture, transfection, and imaging

We prepared organotypic hippocampal slice cultures following previously established protocols (**Chen et al., 2014**). Briefly, slices were obtained from P7-8 Sprague–Dawley rats. At DIV 3, slices were biolistically transfected using a gene gun (Bio-Rad). Gold particles (1.6 μm in diameter; Bio-Rad) were coated with 25 μg of cDNA along with 5 μg of GFP. Live cell imaging was conducted at DIV9/10 using a Nikon confocal system equipped with a water-immersion ×60 objective. Protrusions from dendrites longer than 0.4 μm were counted as spines. For spine density analysis, secondary basal dendrites were selected, and 2–5 different dendrites were imaged from each pyramidal cell. Both image acquisition and spine counting were carried out in a blinded manner.

## Aβ40 and Aβ42 ELISA

We analyzed all cell sample sets using Aβ40 and Aβ42 ELISA kits (Invitrogen, KHB3481 and KHB3441) according to the manufacturer's instructions. The cells were sonicated in RIPA buffer (Solarbio) containing a protease inhibitor cocktail (Selleck), followed by centrifugation at 12,000 × $g$ for 10 minutes to prepare the samples.

## Lentivirus production

We inserted β-CTF-flag and GFP into the pFHTrePW vector (**Ni et al., 2023**) backbone under the control of the Tet Response Element (TRE 3G). The rtTA for the tet-off system was expressed under the control of the human synapsin I promoter. The pFHTrePW plasmid containing β-CTF-flag or GFP, along with PSPAX2 and PMD2g, was transfected into HEK293T cells. After 48 hours, the cell culture medium containing the virus was collected and filtered through a 0.45 μm syringe filter (Millipore). Subsequently, it was concentrated using ultracentrifugation, aliquoted, and stored at –80°C.

## Adeno-associated virus production

We performed AAV purification using triple-transfected HEK293T cells. Briefly, we transfected pHelper, pAAV2-8, and pAAV-MCS containing GFP or β-CTF into HEK293T cells at a ratio of 1:1:1. After 72 hours of incubation, the culture medium was collected in a 50 mL centrifuge tube, and the cell pellet was collected in another tube with lysis buffer (50 mM Tris–HCl, 150 mM NaCl, 2 mM $MgCl_2$, pH 8.0). The cell pellet was subjected to three cycles of freezing and thawing, followed by mixed with the culture medium. PEG8000 (10%) and NaCl (1 M/L) were added to the mixture. After incubating for 1 hour on ice, the mixture was centrifuged for 15 minutes at 12,000 × $g$ and 4°C. The precipitate was gently resuspended in lysis buffer containing DNase I (Roche) using a 1000 ml pipette. After a 30-minute incubation at 37°C, chloroform in a 1:1 (v:v) ratio was added to each tube, shaken for 1 hour, and then centrifuged for 15 minutes at 12,000 × $g$ and 4°C. To harvest the concentrated AAV, the supernatant was processed using an ultrafiltration column (Millipore).

## Electrophysiology

We performed patch-clamp recordings from hippocampal pyramidal cells in acute brain slices. Adult mice approximately 8 weeks old were anesthetized with isoflurane and then decapitated. The hippocampus was harvested intact and placed into cold choline cutting buffer (110 mM choline Cl, 2.5 mM KCl, 1.25 mM NaH$_2$PO$_4$, 25 mM NaHCO$_3$, 7 mM MgSO$_4$, 25 mM D-glucose, 3.1 mM Na pyruvate, 11.6 mM Na ascorbate, 0.5 mM CaCl$_2$ at pH 7.25). Using a vibrating microtome (Leica VT1200S), the hippocampus was sliced into 400 µm sections, followed by a 30-minute recovery period at 34°C and another 30-minute recovery period at room temperature in artificial cerebrospinal fluid (ACSF).

The recording external solution consisted of ACSF containing (in mM) 127 NaCl, 2.5 KCl, 12.5 NaH$_2$PO$_4$, 25 NaHCO$_3$, 25 D-glucose, 2.5 CaCl$_2$, and 1.3 MgCl$_2$, aerated with 95% O$_2$/5% CO$_2$ to maintain a pH around 7.25. Pipettes (TW150F-4, World Precision Instruments) with a resistance of 4–6 mΩ were filled with an internal solution containing (in mM) 115 cesium methanesulfonate, 20 CsCl, 10 HEPES, 2.5 MgCl$_2$, 4 Na$_2$ATP, 0.4 Na$_3$GTP, 10 Na phosphocreatine, and 0.6 EGTA (pH 7.25). mEPSCs were recorded by whole-cell voltage clamp. The mEPSCs were recorded at a holding potential of –70 mV in the presence of 1 µM tetrodotoxin (TTX, MedChem express, HY-12526A) and 100 µM picrotoxin (PTX, Sigma, P1675). EPSCs were analyzed using MiniAnalysis software (Synaptosoft) with an amplitude threshold of 5 pA.

## Antibodies and drugs

The following antibodies were used in this study: Y188 antibody (Abcam, ab32136, RRID:AB_ 2289606), D10E5 antibody (Cell Signaling, 5606, RRID:AB_1903900), Flag antibody (Sigma-Aldrich, F3165, RRID:AB_259529), β-Tubulin III antibody (Sigma, T2200, RRID:AB_262133), GAPDH antibody (Proteintech, 60004-1-Ig, RRID:AB_2107436), HA antibody (Cell Signaling, 3724S, RRID:AB_1549585), GFP antibody (Abcam, ab13970, RRID:AB_300798), SYN antibody (Millipore, AB1543, RRID:AB_2200400), GluR1 antibody (Millipore, MAB2263, RRID:AB_11212678), and GluR2 antibody (Millipore, AB1768-1, RRID:AB_2313802). Peroxidase-conjugated secondary antibodies: goat anti-rabbit IgG (YESEN, 33101ES60, RRID:AB_2922405) and goat anti-mouse IgG (YESEN, 33201ES60, RRID:AB_10015289). Fluorescent secondary antibodies: Alexa 488 goat anti-chicken IgG (Thermo Fisher, A11039, RRID:AB_2534096), Alexa 568 goat anti-chicken IgY (Thermo Fisher, A11041, AB_2534098), Alexa 568 goat anti-rabbit IgG (Thermo Fisher, A11036, AB_10563566), and Alexa 647 goat anti-mouse IgG (Thermo Fisher, A21236, RRID:AB_2535805). Inhibitors: PF-03084014 (Selleck, S7731).

## Plasmids construction

We subcloned the expression of GFP, BACE1-GFP, GFP-Rab5, APP, APP-IRES-BACE1, or related mutations into the pCAGGS mammalian expression vector. The AAV constructs of GFP and β-CTF-HA were subcloned into the pAAV-MCS vector. Lentivirus constructs of GFP, β-CTF, and β-CTF-IRES-GFP were subcloned into the pFHTrePW vector backbone under the control of the TRE 3G.

## Cell lines

HEK293T (ATCC, CRL-11268) was cultured in DMEM supplemented with 10% FBS. PCR testing is performed on the cell culture medium using mycoplasma-specific primers to ensure the cells remain free from mycoplasma contamination every three month.

## Behavioral analysis

Mice (C57BL6/J, Charles River, 213, 5-month-old, male) underwent a 1-hour habituation period in the room before all behavioral tests. All tests were conducted and analyzed in blinded and randomized manners. Sample size was set to be between 13 and 15 based on previous publications focusing on similar behavioral studies (**Ni et al., 2023**). All mice that completed the experiment were included in the analysis.

## Open-field test (OFT)

The mice were introduced into a 40 × 40 cm box without roofs and permitted to explore the apparatus freely for 1 hour. Their activities were monitored using the Ethovision video tracking system (Noldus Information Technology Inc, Leesburg, VA). The field was divided into a center area (24 × 24 cm) and

the whole arena. Parameters such as total travel distance, duration in the center area, velocity, and rearing frequency were automatically recorded and analyzed.

### Y maze

Each mouse was placed in one arm of a Y maze (with arms measuring 30 cm in length) and allowed to explore the maze freely for 8 minutes. The movements of the mice were captured using a video camera positioned above the arena. The number of alternations and entries was analyzed using the Ethovision video tracking system.

### Elevated plus maze

Anxiety-like behavior was assessed using an EPM, which is an elevated apparatus shaped like a plus sign (+). It consists of two open arms (35 × 5 × 0.3 cm), two closed arms (36 × 5 × 18 cm) with 15 cm walls and open roofs, and a 5 × 5 cm central area. At the beginning of the test, each mouse was placed in the center of the maze facing an open arm and allowed to explore the maze freely for 5 minutes. The distance explored and time spent in the apparatus were recorded and analyzed using the SMART software (Panlab, Barcelona, Spain).

### Water T maze

Spatial learning and memory were evaluated using the water T maze behavioral paradigm. In this task, mice were trained to utilize spatial cues within a room to locate a concealed platform and escape from water. In the reversal test, the hidden platform was relocated to the opposite arm to assess cognitive flexibility. The testing apparatus consisted of a plus maze, with each arm measuring 45 cm in length and 10 cm in width, constructed from clear Plexiglas. Each arm was designated as N, S, E, or W. A divider was positioned to block off an arm, allowing the mice to choose only the E or W arm for escape. The water temperature was maintained at 25–26°C and made opaque by adding white, nontoxic powder. An escape platform was submerged 1 cm below the water's surface on the E side of the maze, rendering it invisible to the mice.

At the beginning of each trial, the divider was put in place to block off the appropriate arm. Mice were placed at the starting point and allowed to explore the apparatus freely to reach the platform. If the mouse reached the platform within 1 minute, the response was considered correct; otherwise, it is considered incorrect. The experimenter recorded whether each response was correct or incorrect, and mice were permitted to remain on the platform for 10 seconds before being removed. Each day, mice underwent five trials with starting points semi-randomized between the N and S positions. The criterion for acquisition was achieving 80% or more correct responses averaged across the five trials for two consecutive days. After acquisition training, reversal training commenced. The hidden platform was relocated to the opposite side (W), and the same procedure was repeated until the mice successfully learned the new platform position. Mice that achieved 80% or more correct responses across the five trials for two consecutive days were deemed to have passed the test.

### Fear conditioning

The FC test comprised two components: contextual FC and cued FC. During the training stage, mice were introduced into conditioning boxes and allowed to freely explore the environment for 3 minutes. The baseline freezing behavior was recorded using a visual camera and analyzed with the Ethovision video tracking system. Subsequently, mice were exposed to an 80 dB, 2 kHz tone for 30 seconds (conditioned stimulus), during which an inescapable 0.3 mA foot shock (unconditioned stimulus) was delivered in the last 2 seconds of the tone. This procedure was repeated three times with a 30-second interstimulus interval.

For the contextual FC test, conducted on the second day, mice were placed in the same conditioning box used on the previous day for 5 minutes, and their freezing behavior was recorded for further analysis. In the cued FC test, conducted on the third day after training and contextual FC, mice were placed in a different chamber from the previous one. After a 3-minute free exploration period, mice were exposed to a 3-minute tone stimulus (2 kHz, 80 dB). Freezing behavior during the 3-minute tone stimulus was measured and analyzed using the Ethovision video tracking system.

### Tail suspension test

Each mouse was suspended by its tail using tape affixed to the ceiling of a three-walled rectangular compartment measuring 30 cm in height, 15 cm in width, and 15 cm in depth. The mice dangled

downward, with ample space provided for movement. Video recording was conducted from the side, capturing the animals' immobile time (defined as cessation of limb movements for more than 2 seconds) during the final 4 minutes of a 6-minute session.

## Stereotactic virus injection

Mice aged 4 weeks or 7–8 weeks received injections of lentivirus or AAV in the hippocampus at coordinates ML:±1.5 mm, AP: –2 mm, DV: –1.5 mm. For spine density analysis, mice were harvested after 2 months; for behavioral tests, mice were evaluated after 4 months.

## Liquid chromatography tandem mass spectrometry (LC-MS/MS) sample preparation and analysis

For protein sample preparation, neurons seeded in 24-well plates underwent lysis using ultrasonic waves in RIPA buffer. After centrifugation at 18,000 × $g$ for 10 minutes at 4°C, the supernatants were collected and quantified using a BCA assay (Thermo Fisher). Protein extracts were subjected to an in-solution digest protocol, involving reduction with 5 mM Tris(2-carboxyethyl) phosphine hydrochloride (Aldrich, USA) followed by alkylation with 10 mM iodoacetamide (Sigma, USA). Trypsin was added at a 1:100 ratio, and the mixture was incubated at 37°C overnight in the dark. Digested peptides were then collected by centrifugation and desalted using C18 tips (Pierce, USA) for subsequent analysis.

For LC-MS/MS analysis, the peptide mixture was analyzed using an online EASY-nLC 1000 HPLC system coupled with an Orbitrap Fusion mass spectrometer (Thermo Fisher Scientific). The sample was loaded directly onto a 15 cm homemade capillary column (100 μm I.D., C18-AQ 1.9 μm resin, Dr. Maisch). Mobile phase A consisted of 0.1% formic acid (FA), 2% acetonitrile (ACN), and 98% $H_2O$, while mobile phase B comprised 0.1% FA, 2% $H_2O$, and 98% ACN. A 180-minute gradient (mobile phase B: 2% at 0 minute, 5% at 7 minutes, 20% at 127 minutes, 35% at 167 minutes, 95% at 173 minutes, 95% at 180 minutes) was employed at a static flow rate of 300 nL/minute.

We acquired data for proteomic analysis in a data-dependent mode, starting with one full MS1 scan in the Orbitrap (m/z: 300–1800; resolution: 120,000; AGC target value: 500,000; maximal injection time: 50 ms), followed by an MS2 scan in the linear ion trap (32% normalized collision energy; maximal injection time: 250 ms). The isolation window was set at 1.6 m/z.

## Statistics

Data were processed using Microsoft Excel, and statistical analysis was performed with GraphPad Prism 7. Figures were made using Adobe Illustrator V26.3.1. Sample sizes and the statistical analyses performed are described in the respective figure legends. For all analyses, all experimental replicates were biological replicates, and $p < 0.05$ was considered statistically significant.

## Acknowledgements

We thank the staff members of the Animal Facility at the National Facility for Protein Science in Shanghai (NFPS), Shanghai Advanced Research Institute, Chinese Academy of Sciences, China, for providing assistance in mouse breeding and maintenance. This study was supported by the Shanghai Basic Research Pioneer Project.

## Additional information

### Funding

| Funder | Grant reference number | Author |
| --- | --- | --- |
| National Natural Science Foundation of China | 31671044 | Yelin Chen |
| National Natural Science Foundation of China | 91849204 | Yelin Chen |
| National Program on Key Research Project of China | 2016YFA0501901 | Yelin Chen |

| Funder | Grant reference number | Author |
|---|---|---|
| Shanghai Municipal Science and Technology Major Project | 2019SHZDZX02 | Yelin Chen |
| Shanghai Basic Research Pioneer Project | | Yelin Chen |

The funders had no role in study design, data collection and interpretation, or the decision to submit the work for publication.

## Author contributions

Mengxun Luo, Conceptualization, Data curation, Formal analysis, Validation, Investigation, Visualization, Methodology, Writing – original draft; Jia Zhou, Cailu Sun, Formal analysis, Validation, Investigation; Wanjia Chen, Chaoying Fu, Validation, Investigation; Chenfang Si, Data curation, Formal analysis, Validation, Investigation; Yaoyang Zhang, Data curation; Yang Geng, Conceptualization, Methodology, Writing – review and editing; Yelin Chen, Conceptualization, Supervision, Funding acquisition, Methodology, Project administration, Writing – review and editing

## Author ORCIDs

Yaoyang Zhang ⓘ https://orcid.org/0000-0001-5363-9834
Yelin Chen ⓘ https://orcid.org/0000-0002-9793-2941

## Ethics

All animal procedures received approval from the Institutional Animal Care and Use Committee of Animal Facility at the National Facility for Protein Science in Shanghai (NFPS), Shanghai Advanced Research Institute, Chinese Academy of Sciences, China, and adhered to the National Institutes of Health's Guide for the Care and Use of Laboratory Animals. The reference number of the approvals of the animal study is IACUC 202003080007.

Reviewer #1 (Public review): https://doi.org/10.7554/eLife.100968.3.sa1
Reviewer #3 (Public review): https://doi.org/10.7554/eLife.100968.3.sa2
Author response https://doi.org/10.7554/eLife.100968.3.sa3

# Additional files

## Supplementary files

MDAR checklist

## Data availability

All data generated or analysed during this study are included in the manuscript and supporting files; source data files have been provided for all figures.

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
