## [Editor Report · eLife Assessment]

This study presents a **useful** demonstration that a specific protein fragment may induce the loss of synapses in Alzheimer's disease. The evidence supporting the data is **solid** but only partially supports the conclusion and would benefit from additional discussion indicated by the literature from reviewer #1. The application of the findings is limited because blocking the formation of the protein fragment has not benefited patients in several clinical trials.

---

## [Referee Report · Reviewer #1 (Public review)]

Summary of what the authors were trying to achieve:

In this manuscript, the authors investigated the role of β-CTF on synaptic function and memory. They report that β-CTF can trigger the loss of synapses in neurons that were transiently transfected in cultured hippocampal slices and that this synapse loss occurs independently of Aβ. They confirmed previous research (Kim et al, Molecular Psychiatry, 2016) that β-CTF-induced cellular toxicity occurs through a mechanism involving a hexapeptide domain (YENPTY) in β-CTF that induces endosomal dysfunction. Although the current study also explores the role of β-CTF in synaptic and memory function in the brain using mice chronically expressing β-CTF, the studies are inconclusive because potential effects of Aβ generated by γ-secretase cleavage of β-CTF were not considered. Based on their findings, the authors suggest developing therapies to treat Alzheimer's disease by targeting β-CTF. While they acknowledge that clinical trials of potent BACE1 inhibitors - which also target β-CTF - have failed to show clinical improvement, their study lacks in vivo evidence directly linking β-CTF to brain function, which weakens its significance.

Major strengths and weaknesses of the methods and results:

The conclusions of the in vitro experiments using cultured hippocampal slices were well supported by the data, but aspects of the in vivo experiments need additional clarification.

In contrast to the in vitro experiments in which a γ-secretase inhibitor was used to exclude possible effects of Aβ, this possibility was not examined in in vivo experiments assessing synapse loss and function (Fig. 3) and cognitive function (Fig. 4). The absence of plaque formation (Fig. 4C) is not sufficient to exclude the possibility that Aβ is involved. The potential involvement of Aβ is an important consideration given the 4-month duration of protein expression in the in vivo studies. This issue could be addressed using γ-secretase modulators to avoid the off-target effects of inhibitors. Evidence that the detrimental effects in mice are directly caused by β-CTF rather than indirectly via Aβ is critical to support the authors' conclusion.

Appraisal of whether the authors achieved their aims, and whether the results support their conclusion:

See above

Discussion of likely impact of the work on the field, and the utility of the methods and data to the community:

The authors' use of sparse expression to examine the role of β-CTF on spine loss could be a useful general tool for examining synapses in brain tissue.

Any additional context that might help readers interpret or understand the significance of the work:

The discovery of BACE1 stimulated an international effort to develop BACE1 inhibitors to treat Alzheimer's disease. BACE1 inhibitors block the formation of β-CTF which, in turn, prevents the formation of Aβ and other fragments. Unfortunately, BACE1 inhibitors not only did not improve cognition in patients with Alzheimer's disease, they appeared to worsen it, suggesting that β-CTF could facilitate learning and memory. Therefore, it seems unlikely that the disruptive effects of β-CTF on endosomes plays a significant role in the human disease.

Comments on revisions:

The authors may be interested in the study by Ma et al., PNAS 2007 titled "Involvement of β-site APP cleaving enzyme 1 (BACE1) in amyloid precursor protein-mediated enhancement of memory and activity-dependent synaptic plasticity," which provides significant insights into the physiological role of BACE1 in synaptic function. The researchers demonstrated that BACE1-mediated cleavage of amyloid precursor protein (APP) is essential for enhancing learning, memory, and synaptic plasticity in vivo. They observed that overexpression of APP in transgenic mice led to improved spatial memory retention and potentiation of synaptic plasticity, effects that were abolished when one or both copies of the BACE1 gene were eliminated. This suggests that BACE1's cleavage of APP facilitates activity-dependent synaptic modifications, potentially through the production of APP intracellular domain (AICD) via β-CTF, rather than amyloid-β (Aβ) or soluble APPα (sAPPα). These findings highlight a physiological mechanism where BACE1-mediated APP processing leading to β-CTF supports cognitive functions, potentially explaining the detrimental effects of BACE1 inhibitors on cognitive function in clinical trials.

---

## [Referee Report · Reviewer #3 (Public review)]

Summary:

Most previous studies have focused on the contributions of Abeta and amyloid plaques in the neuronal degeneration associated with Alzheimer's disease, especially in the context of impaired synaptic transmission and plasticity which underlies the impaired cognitive functions, a hallmark in AD. But processes independent of Abeta and plaques are much less explored, and to some extent, the contributions of these processes are less well understood. Luo et all addressed this important question with an array of approaches, and their findings generally support the contribution of beta-CTF-dependent but non-Abeta dependent process to the impaired synaptic properties in the neurons. Interestingly, the above process appears to operate in a cell-autonomous manner. This cell-autonomous effect of beta-CTF as reported here may facilitate our understanding of some potential important cellular processes related to neurodegeneration. Although these findings are valuable, it is key to understand the probability of this process occurring in a more natural condition, such as when this process occurring in many neurons at the same time. This will put the authors' findings into a context for a better understanding of their contribution to either physiological or pathological processes, such as Alzheimer's. The experiments and results using cell system are quite solid, but the in vivo results are incomplete and hence less convincing (see below). The mechanistic analysis is interesting but primitive, and does not add much more weight to the significance. Hence, further efforts from the authors are required to clarify, and solidify their results, in order to provide a complete picture and support for the authors' conclusions.

Strengths:

(1) The authors have addressed an interesting and potentially important question

(2) The analysis using the cell system are solid and provides strong support for the authors' major conclusions. This analysis has used various technical approaches to support the authors' conclusions from different aspects and most of these results are consistent with each other.

Weaknesses:

(1) The relevance of the authors' major findings to the pathology, especially the Abeta-dependent processes is less clear, and hence the importance of these findings may be limited.

(2) In vivo analysis is incomplete, with certain caveats in the experimental procedures and some of the results need to be further explored to confirm the findings.

(3) The mechanistic analysis is rather primitive and does not add further significance.

Comments on revisions:

The authors have satisfactorily addressed my main questions.

---

## [Author Response]

The following is the authors’ response to the original reviews.

**Public Reviews:**

**Reviewer #1 (Public Review):**
Summary of what the authors were trying to achieve:In this manuscript, the authors investigated the role of β-CTF on synaptic function and memory. They report that β-CTF can trigger the loss of synapses in neurons that were transiently transfected in cultured hippocampal slices and that this synapse loss occurs independently of Aβ. They confirmed previous research (Kim et al, Molecular Psychiatry, 2016) that β-CTF-induced cellular toxicity occurs through a mechanism involving a hexapeptide domain (YENPTY) in β-CTF that induces endosomal dysfunction. Although the current study also explores the role of β-CTF in synaptic and memory function in the brain using mice chronically expressing β-CTF, the studies are inconclusive because potential effects of Aβ generated by γ-secretase cleavage of β-CTF were not considered. Based on their findings, the authors suggest developing therapies to treat Alzheimer's disease by targeting β-CTF, but did not address the lack of clinical improvement in trials of several different BACE1 inhibitors, which target β-CTF by preventing its formation.

We would like to thank the reviewer for his/her suggestions. We have addressed the specific comments in following sections.

Major strengths and weaknesses of the methods and results:The conclusions of the in vitro experiments using cultured hippocampal slices were well supported by the data, but aspects of the in vivo experiments and proteomic studies need additional clarification.(1) In contrast to the in vitro experiments in which a γ-secretase inhibitor was used to exclude possible effects of Aβ, this possibility was not examined in in-vivo experiments assessing synapse loss and function (Figure 3) and cognitive function (Figure 4). The absence of plaque formation (Figure 4B) is not sufficient to exclude the possibility that Aβ is involved. The potential involvement of Aβ is an important consideration given the 4-month duration of protein expression in the in vivo studies.

We appreciate the reviewer for raising this question. While our current data did not exclude the potential involvement of Aβ-induced toxicity in the synaptic and cognitive dysfunction observed in mice overexpressing β-CTF, addressing this directly remains challenging. Treatment with γ-secretase inhibitors could potentially shed light on this issue. However, treatments with γ-secretase inhibitors are known to lead to brain dysfunction by itself likely due to its blockade of the γ-cleavage of other essential molecules, such as Notch[1, 2]. Therefore, this approach is unlikely to provide a clear answer, which prevents us from pursuing it further experimentally in vivo. We hope the reviewer understands this limitation. We have included additional discussion (page 14 of the revised manuscript) to highlight this question.

(2) The possibility that the results of the proteomic studies conducted in primary cultured hippocampal neurons depend in part on Aβ was also not taken into consideration.

We thank the reviewer for raising this question. In the revised manuscript, we examined the protein levels of synaptic proteins after treatment with γ-secretase inhibitors and found that the levels of certain synaptic proteins were further reduced in neurons expressing β-CTF (Supplementary figure 5A-B). These results do not support Aβ as a major contributor of the proteomic changes induced by β-CTF.

Likely impact of the work on the field, and the utility of the methods and data to the community:

The authors' use of sparse expression to examine the role of β-CTF on spine loss could be a useful general tool for examining synapses in brain tissue.

We thank the reviewer for these comments.

Additional context that might help readers interpret or understand the significance of the work:The discovery of BACE1 stimulated an international effort to develop BACE1 inhibitors to treat Alzheimer's disease. BACE1 inhibitors block the formation of β-CTF which, in turn, prevents the formation of Aβ and other fragments. Unfortunately, BACE1 inhibitors not only did not improve cognition in patients with Alzheimer's disease, they appeared to worsen it, suggesting that producing β-CTF actually facilitates learning and memory. Therefore, it seems unlikely that the disruptive effects of β-CTF on endosomes plays a significant role in human disease. Insights from the authors that shed further light on this issue would be welcome.

Response: We would like to express our gratitude to the reviewer for raising this question. It remains puzzling why BACE1 inhibition has failed to yield benefits in AD patients, while amyloid clearance via Aβ antibodies are able to slow down disease progression. One possible explanation is that pharmacological inhibition of BACE1 may not be as effective as its genetic removal. Indeed, genetic depletion of BACE1 leads to the clearance of existing amyloid plaques[3], whereas its pharmacological inhibition prevents the formation of new plaques but does not deplete the existing ones[4]. We think the negative results of BACE1 inhibitors in clinical trials may not be sufficient to rule out the potential contribution of β-CTF to AD pathogenesis. Given that cognitive function continues to deteriorate rapidly in plaque-free patients after 1.5 years of treatment with Aβ antibodies in phase three clinical studies[5], it is important to consider the potential role of other Aβ-related fragments in AD pathogenesis, such as β-CTF. We included further discussion in the revised manuscript (page 15 of the revised manuscript) to discusss this question.

**Reviewer #2 (Public Review):**
Summary:In this study, the authors investigate the potential role of other cleavage products of amyloid precursor protein (APP) in neurodegeneration. They combine in vitro and in vivo experiments, revealing that β-CTF, a product cleaved by BACE1, promotes synaptic loss independently of Aβ. Furthermore, they suggest that β-CTF may interact with Rab5, leading to endosomal dysfunction and contributing to the loss of synaptic proteins.

We would like to thank the reviewer for his/her suggestions. We have addressed the specific comments in following sections.

Weaknesses:Most experiments were conducted in vitro using overexpressed β-CTF. Additionally, the study does not elucidate the mechanisms by which β-CTF disrupts endosomal function and induces synaptic degeneration.

We would like to thank the reviewer for this comment. While a significant portion of our experiments were conducted in vitro, the main findings were also confirmed in vivo (Figure 3 and 4). Repeating all the experiments in vivo would be challenging and may not be possible because of technical difficulties. Regarding the use of overexpressed β-CTF, we acknowledge that this represents a common limitation in neurodegenerative disease studies. These diseases progress slowly over decades in patients. To model this progression in cell or mouse models within a time frame feasible for research, overexpression of certain proteins is often inevitable. Since β-CTF levels are elevated in AD patients[6], its overexpression is not a irrelevant approach to investigate its potential effects.

We did not further investigate the mechanisms by which β-CTF disrupted endosomal function because our preliminary results align with previous findings that could explain its mechanism. Kim et al. demonstrated that β-CTF recruits APPL1 (a Rab5 effector) via the YENPTY motif to Rab5 endosomes, where it stabilizes active GTP-Rab5, leading to pathologically accelerated endocytosis, endosome swelling and selectively impaired transport of Rab5 endosomes[6]. However, this paper did not show whether this Rab5 overactivation-induced endosomal dysfunction leads to any damages in synapses. In our study, we observed that co-expression of Rab5_S34N_ with β-CTF effectively mitigated β-CTF-induced spine loss in hippocampal slice cultures (Figures 6L-M), indicating that Rab5 overactivation-induced endosomal dysfunction contributed to β-CTF-induced spine loss. We included further discussion in the revised manuscript to clarify this (page 15 of the revised manuscript).

**Reviewer #3 (Public Review):**
Summary:Most previous studies have focused on the contributions of Abeta and amyloid plaques in the neuronal degeneration associated with Alzheimer's disease, especially in the context of impaired synaptic transmission and plasticity which underlies the impaired cognitive functions, a hallmark in AD. But processes independent of Abeta and plaques are much less explored, and to some extent, the contributions of these processes are less well understood. Luo et all addressed this important question with an array of approaches, and their findings generally support the contribution of beta-CTF-dependent but non-Abeta-dependent process to the impaired synaptic properties in the neurons. Interestingly, the above process appears to operate in a cell-autonomous manner. This cell-autonomous effect of beta-CTF as reported here may facilitate our understanding of some potentially important cellular processes related to neurodegeneration. Although these findings are valuable, it is key to understand the probability of this process occurring in a more natural condition, such as when this process occurs in many neurons at the same time. This will put the authors' findings into a context for a better understanding of their contribution to either physiological or pathological processes, such as Alzheimer's. The experiments and results using the cell system are quite solid, but the in vivo results are incomplete and hence less convincing (see below). The mechanistic analysis is interesting but primitive and does not add much more weight to the significance. Hence, further efforts from the authors are required to clarify and solidify their results, in order to provide a complete picture and support for the authors' conclusions.

We would like to thank the reviewer for the suggestions. We have addressed the specific comments in following sections.

Strengths:(1) The authors have addressed an interesting and potentially important question(2) The analysis using the cell system is solid and provides strong support for the authors' major conclusions. This analysis has used various technical approaches to support the authors' conclusions from different aspects and most of these results are consistent with each other.

We would like to thank the reviewer for these comments.

Weaknesses:(1) The relevance of the authors' major findings to the pathology, especially the Abeta-dependent processes is less clear, and hence the importance of these findings may be limited.

We would like to thank the reviewer for this question. Phase 3 clinical trial data from Aβ antibodies show that cognitive function continues to decline rapidly, even in plaque-free patients, after 1.5 years of treatment[5]. This suggests that plaque-independent mechanisms may drive AD progression. Therefore, it is crucial to consider the potential contributions of other Aβ species or related fragments, such as alternative forms of Aβ and β-CTF. While it is early to predict how much β-CTF contributes to AD progression, it is notable that β-CTF induced synaptic deficits in mice, which recapitulates a key pathological feature of AD. Ultimately, the contribution of β-CTF in AD pathogenesis can only be tested through clinical studies in the future.

(2) In vivo analysis is incomplete, with certain caveats in the experimental procedures and some of the results need to be further explored to confirm the findings.

We would like to thank the reviewer for this suggestion. We have corrected these caveats in the revised manuscript.

(3) The mechanistic analysis is rather primitive and does not add further significance.

We would like to thank the reviewer for this comment. We did not delve further into the underlying mechanisms because our analysis indicates that Rab5 overactivation-induced endosomal dysfunction underlies β-CTF-induced synaptic dysfunction, which is consistent with another study and has been addressed in our study[6]. We hope the reviewer could understand that our focus in this paper is on how β-CTF triggers synaptic deficits, which is why we did not investigate the mechanisms of β-CTF-induced endosomal dysfunction further.

**Recommendations for the authors:**

**Reviewer #1 (Recommendations For The Authors):**
Suggestions for improved or additional experiments, data, or analyses:(1) In Figures 4H, 4J, 4K and Supplemental Figures 3C, 3E, and 3G, it was unclear whether a repeated measures 2-way ANOVA, rather than a 2-way ANOVA, followed by appropriate post-hoc analyses was used to strengthen the conclusion that there were significant effects in the behavioral tests.

We appreciate the reviewer for raising this point and apologize for the lack of clear description in the manuscript. In those figures mentioned above, we use a repeated measures 2-way ANOVA to analyze the data by Graphpad Prism. In Figure 4H, fear conditioning tests were conducted. The same cohort of mice were used in the baseline, contextual and cued tests. Firstly, baseline freezing was tested; then these mice underwent tone and foot shock training, followed by contextual test and cued test. So, a repeated measures 2-way ANOVA is more appropriate for the experiment.

In water T maze tests (Figure 4J and K), the same cohort of mice were trained and tested each day. So, it’s also appropriate to use a repeated measures 2-way ANOVA.

In Supplementary figure 3C, 3E and 3G, OFT was conducted. In this experiment, the locomotion of the same cohort of mice were recorded. Also, it’s appropriate to use a repeated measures 2-way ANOVA.

Clearer description for these experiments has been provided in the revised manuscript.

(2) Including gender analyses would be helpful.

The mice we used in this study were all males.

Minor corrections to text and figures:(1) Quantitative analyses in Figures 5A-C, 5H, 6G, 6H, and Supplementary Figures 4 and 5C would be helpful.

We have provided quantitative analysis of these results (Figure 5D, 5J, 6K, Supplementary figure 4D, 5F) mentioned above in the revised manuscript.

(2) Percent correct (%) in Figures 4J and 4K should be labeled as 0, 50, and 100 instead of 0.0, 0.5, and 1.0.

We would like to thank the reviewer for pointing out this. We have made corrections in the revised manuscript.

**Reviewer #2 (Recommendations For The Authors):**
In the study conducted by Luo et al, it was observed that the fragment of amyloid precursor protein (APP) cleaved by beta-site amyloid precursor protein cleaving enzyme 1 (BACE1), known as β-CTF, plays a crucial role in synaptic damage. The study found increasing expression of β-CTF in neurons could induce synapse loss both in vitro and in vivo, independent of Aβ. Mechanistically, they explored how β-CTF could interfere with the endosome system by interacting with RAB5. While this study is intriguing, there are several points that warrant further investigation:(1) The study involved overexpressing β-CTF in neurons. It would be valuable to know if the levels of β-CTF are similarly increased in Alzheimer's disease (AD) patients or AD mouse models.

We would like to thank the reviewer for the suggestion. It’s reported β-CTF levels were significantly elevated in the AD cerebral cortex[6]. Most AD mouse models are human APP transgenic mouse models with elevated β-CTF levels[7].

(2) The study noted that β-CTF in neurons is a membranal fragment, but the overexpressed β-CTF was not located in the membrane. It is important to ascertain whether the membranal β-CTF and cytoplasmic β-CTF lead to synapse loss in a similar manner.

We apologize for not clearly explaining the localization of β-CTF in the original manuscript. β-CTF is produced from APP through β-cleavage, a process that occurs in organelles such as endo-lysosomes[8]. The overexpressed β-CTF is also primarily localized in the endo-lysosomal systems (Figure 5C and Supplementary figure 4C), similar to those generated by APP cleavage.

(3) The study found a significant decrease in GluA1, a subunit of AMPA receptors, due to β-CTF. It would be beneficial to investigate whether there are systematic alterations in NMDA receptors, including GluN2A and GluN2B.

We would like to express our gratitude to the reviewer for bringing up this question. The protein levels of GluN2A and GluN2B are also reduced in neurons expressing β-CTF (Figure 6E-F)

(4) The study showed a significant decrease in the frequency of miniature excitatory postsynaptic currents (mEPSC), indicating disrupted presynaptic vesicle neurotransmitter release. It would be pertinent to test whether the expression level of the presynaptic SNARE complex, which is required for vesicle release, is altered by β-CTF.

We would like to express our gratitude to the reviewer for bringing up this question. The protein level of the presynaptic SNARE complex, such as VAMP2, is also reduced in neurons expressing β-CTF (Figure 6E, G).

(5) Since AMPA receptors are glutamate receptors, it is important to determine whether the ability of glutamate release is altered by β-CTF. In vivo studies using a glutamate sensor should be conducted to examine glutamate release.

We would like to express our gratitude to the reviewer for this suggestion. It will be interesting to use glutamate sensors to assess the ability of glutamate release in the future.

(6) The quality of immunostaining associated with Figures 4B and 4C was noted to be suboptimal.

We apologize for the suboptimal quality of these images. The immunostaining in Figures 4B and 4C were captured using the stitching function of a confocal microscope to display larger areas, including the entire hemisphere and hippocampus. We have reprocessed the images to obtain higher-quality versions.

(7) It would be insightful to investigate whether treatment with a BACE1 inhibitor in the study could reverse synaptic deficits mediated by β-CTF.

We would like to thank the reviewer for this sggestion. In Figure 1I-M, we constructed an APP mutant (APP_MV_), which cannot be cleaved by BACE1 to produce β-CTF and Aβ but has no impact on β’-cleavage. When co-expressed with BACE1, APP_MV_ failed to induce spine loss, supporting the effect of β-CTF. We think these results domonstrate that β-CTF underlies the synaptic deficits. It would be interesting to test the effects of BACE1 inhibition in the future.

(8) Considering the potential implications for therapeutics, it is worth exploring whether extremely low levels of β-CTF have beneficial effects in regulating synaptic function or promoting synaptogenesis at a physiological level.

We would like to thank the reviewer for raising this question. We found that when the plasmid amount was reduced to 1/8 of the original dose, β-CTF no longer induced a decrease in dendritic spine density (Supplementary figure 2E-F). It’s reported APP-Swedish mutation in familial AD increased synapse numbers and synaptic transmission, whereas inhibition of BACE1 lowered synapse numbers, suppressed synaptic transmission in wild type neurons, suggesting that at physiological level, β-CTF might be synaptogenic[9].

(9) The molecular mechanism through which β-CTF interferes with Rab5 function should be elucidated.

We would like to thank the reviewer for raising this question. Kim et al have elucidated the mechanism through which β-CTF interferes with Rab5 function. β-CTF recruited APPL1 (a Rab5 effector) via YENPTY motif to Rab5 endosomes, where it stabilizes active GTP-Rab5, leading to pathologically accelerated endocytosis, endosome swelling and selectively impaired transport of Rab5 endosomes[6]. We have included additional discussion for this question in the revised manuscript (page 15 of the revised manuscript).

(10) The study could compare the role of β-CTF and Aβ in neurodegeneration in AD mouse models.

We would like to thank the reviewer for raising this point. While it is easier to dissect the role of Aβ and β-CTF in vitro, some of the critical tools are not applicabe in vivo, such as γ-secretase inhibitors, which lead to severe side effects because of their inhibition on other γ substrates[1, 2]. Therefore it will be difficult to deomonstrate their different roles in vivo. There are studies showing that β-CTF accumulation precedes Aβ deposition in model mice and mediates Aβ independent intracellular pathologies[10, 11], consistent with our results.

(11) Based on the findings, it would be valuable to discuss possible explanations for the failure of most BACE1 inhibitors in recent clinical trials for humans.

Response: We would like to express our gratitude to the reviewer for raising this recommendation. It is a big puzzle why BACE1 inhibition failed to provide beneficial effects in AD patients whereas clearance of amyloid by Aβ antibodies could slow down the AD progress. One potential answer is that pharmacological inhibition of BACE1 might be not as effective as its genetic removal. Indeed, genetic depletion of BACE1 leads to clearance of existing amyloid plaques[3], whereas pharmacological inhibition of BACE1 could not stop growth of existing plaques, although it prevents formation of new plaques[4]. The negative result of BACE1 inhibitors might not be sufficient to exclude the possibility that β-CTF could also contribute to the AD pathogenesis. We have included additional discussion for this question in the revised manuscript (page 15 of the revised manuscript).

**Reviewer #3 (Recommendations For The Authors):**
Major:(1) The cell experiments were performed at DIV 9, do the authors know whether at this age, the neurons are still developing and spine density has not reached a pleated yet? If so, the observed effect may reflect the impact on development and/or maturation, rather than on the mature neurons. The authors should be more specific about this issue.

We would like to thank the reviewer for pointing out this question. These slice cultures were made from 1-week-old rats. DIV 9 is about two weeks old. These neurons are still developing and spine density has not reached a plateau yet[12]. In addition, we also investigated the effects of β-CTF on the synapses of mature neurons in two-month-old mice (Figure 3). So we think the observed effect reflects the impact on both immature and mature neurons.

(2) mEPSCs shown in Figure 3D were of small amplitudes, perhaps also indicating that these synapses are not yet mature.

In Figure 3D, the mEPSC results were obtained from pyramidal neurons in the CA1 region of two-month-old mice. At the age of two months, neurotransmitter levels and synaptic density have reached adult levels[13].

(3) There was no data on the spine density or mEPSCs in the mice OE b-CTF, hence it is unclear whether a primary impact of this manipulation (b-CTF effect) on the synaptic transmission still occurs in vivo.

In Figure 3, we examined the density of dendritic spines and mEPSCs from CA1 pyramidal neurons infected with lentivirus expressing β-CTF in mice and showed that those neurons expressing additional amount of β-CTF exhibited lower spine density and less mEPSCs, supporting that β-CTF also damaged synaptic transmission in vivo.

(4) OE of b-CTF should lead to the production of Abeta, although this may not lead to the formation of significant plaques. How do the authors know whether their findings on behavioral and cognitive impairments were not largely mediated by Abeta, which has been widely reported by previous studies?

We would like to thank the reviewer for pointing out this question. Indeed, our in vivo data could not exclude the potential involvement of Aβ in the pathology, despite the absence of amyloid plaque formation. It will be difficult to demonstrate this question in vivo because of the severe side effects from γ inhibition.

(5) Figure 4H, the freezing level in the cued fear conditioning was very high, likely saturated; this may mask a potential reduction in the b-CTF OE mice (there is a hint for that in the results). The authors should repeat the experiments using less strong footshock strength (hence resulting in less freezing, <70%).

We would like to express our gratitude to the reviewer for bringing up this question. The contextual fear conditioning test assesses hippocampal function, while the cued fear conditioning test assesses amygdala function. We hope the reviewer understands that our primary goal is to assess hippocampus-related functions in this experiment and we did see a significant difference between GFP and β-CTF groups. Therefore, we think the intensity of footshock we used was suitable to serve the primary purpose of this experiment.

(6) Why was the deficit in the Morris water maze in the b-CTF OE mice only significant in the training phase?

We would like to thank the reviewer for rasing this question and apologize for not describing the test clearly. This is a water T maze test, not Morris water maze test.

To make the behavioral paradigm of the water T maze test easier to understand, we have provided a more detailed description of the methods in the new version of the manuscript.

The acquisition phase of the Water T Maze (WTM) evaluates spatial learning and memory, where mice use spatial cues in the environment to navigate to a hidden platform and escape from water, while the reversal learning measures cognitive flexibility in which mice must learn a new location of the hidden platform[14]. In reversal learning task (Figure 4J-K), the learning curves of the two groups of mice did not show any significant differences, indicating that the expression of β-CTF only damages spatial learning and memory but not cognitive flexibility. This is consistent with a previous report using APP/PS1 mice[15].

(7) Will the altered Rab5 in the b-CTF OE condition also affect the level of other proteins?

We would like to express our gratitude to the reviewer for raising this interesting question. Expression of Rab5_S34N_ in β-CTF-expressing neurons did not alter the levels of synapse-related proteins that were reduced in these neurons (Supplementary figure 5G-H), suggesting Rab5 overactivation did not contribute to these protein expression changes induced by β-CTF.

(8) How do the authors reconcile their findings with the well-established findings that Abeta affects synaptic transmission and spine density? Do they think these two processes may occur simultaneously in the neurons, or, one process may dominate in the other?

APP, Aβ, and presenilins have been extensively studied in mouse models, providing convincing evidence that high Aβ concentrations are toxic to synapses[16]. Moreover, addition of Aβ to murine cultured neurons or brain slices is toxic to synapses[17]. However, Aβ-induced synaptotoxicity was not observed in our study. A major difference between our study and others is that our study used a isolated expression system that apply Aβ only to individual neurons surrounded by neurons without excessive amount of Aβ, whereas the rest studies generally apply Aβ to all the neurons. Therefore, we predict that Aβ does not lead to synaptic deficits from individual neurons in cell autonomous manners, whereas β-CTF does. Aβ and β-CTF represent two parallel pathways of action. Additional discussion for this question has been included in the revised manuscript (page 14 of the revised manuscript).

Minor:Fig 2F-G, "prevent" rather than "reverse"?

We would like to thank the reviewer for pointing this out. We have made corrections in the revised manuscript.

Reference:

(1) GüNER G, LICHTENTHALER S F. The substrate repertoire of γ-secretase/presenilin [J]. Seminars in cell & developmental biology, 2020, 105: 27-42.

(2) DOODY R S, RAMAN R, FARLOW M, et al. A phase 3 trial of semagacestat for treatment of Alzheimer's disease [J]. The New England journal of medicine, 2013, 369(4): 341-50.

(3) HU X, DAS B, HOU H, et al. BACE1 deletion in the adult mouse reverses preformed amyloid deposition and improves cognitive functions [J]. The Journal of experimental medicine, 2018, 215(3): 927-40.

(4) PETERS F, SALIHOGLU H, RODRIGUES E, et al. BACE1 inhibition more effectively suppresses initiation than progression of β-amyloid pathology [J]. Acta neuropathologica, 2018, 135(5): 695-710.

(5) SIMS J R, ZIMMER J A, EVANS C D, et al. Donanemab in Early Symptomatic Alzheimer Disease: The TRAILBLAZER-ALZ 2 Randomized Clinical Trial [J]. Jama, 2023, 330(6): 512-27.

(6) KIM S, SATO Y, MOHAN P S, et al. Evidence that the rab5 effector APPL1 mediates APP-βCTF-induced dysfunction of endosomes in Down syndrome and Alzheimer's disease [J]. Molecular psychiatry, 2016, 21(5): 707-16.

(7) MONDRAGóN-RODRíGUEZ S, GU N, MANSEAU F, et al. Alzheimer's Transgenic Model Is Characterized by Very Early Brain Network Alterations and β-CTF Fragment Accumulation: Reversal by β-Secretase Inhibition [J]. Frontiers in cellular neuroscience, 2018, 12: 121.

(8) ZHANG X, SONG W. The role of APP and BACE1 trafficking in APP processing and amyloid-β generation [J]. Alzheimer's research & therapy, 2013, 5(5): 46.

(9) ZHOU B, LU J G, SIDDU A, et al. Synaptogenic effect of APP-Swedish mutation in familial Alzheimer's disease [J]. Science translational medicine, 2022, 14(667): eabn9380.

(10) LAURITZEN I, PARDOSSI-PIQUARD R, BAUER C, et al. The β-secretase-derived C-terminal fragment of βAPP, C99, but not Aβ, is a key contributor to early intraneuronal lesions in triple-transgenic mouse hippocampus [J]. The Journal of neuroscience : the official journal of the Society for Neuroscience, 2012, 32(46): 16243-1655a.

(11) KAUR G, PAWLIK M, GANDY S E, et al. Lysosomal dysfunction in the brain of a mouse model with intraneuronal accumulation of carboxyl terminal fragments of the amyloid precursor protein [J]. Molecular psychiatry, 2017, 22(7): 981-9.

(12) HARRIS K M, JENSEN F E, TSAO B. Three-dimensional structure of dendritic spines and synapses in rat hippocampus (CA1) at postnatal day 15 and adult ages: implications for the maturation of synaptic physiology and long-term potentiation [J]. The Journal of neuroscience : the official journal of the Society for Neuroscience, 1992, 12(7): 2685-705.

(13) SEMPLE B D, BLOMGREN K, GIMLIN K, et al. Brain development in rodents and humans: Identifying benchmarks of maturation and vulnerability to injury across species [J]. Progress in neurobiology, 2013, 106-107: 1-16.

(14) GUARIGLIA S R, CHADMAN K K. Water T-maze: a useful assay for determination of repetitive behaviors in mice [J]. Journal of neuroscience methods, 2013, 220(1): 24-9.

(15) ZOU C, MIFFLIN L, HU Z, et al. Reduction of mNAT1/hNAT2 Contributes to Cerebral Endothelial Necroptosis and Aβ Accumulation in Alzheimer's Disease [J]. Cell reports, 2020, 33(10): 108447.

(16) CHAPMAN P F, WHITE G L, JONES M W, et al. Impaired synaptic plasticity and learning in aged amyloid precursor protein transgenic mice [J]. Nature neuroscience, 1999, 2(3): 271-6.

(17) WANG Z, JACKSON R J, HONG W, et al. Human Brain-Derived Aβ Oligomers Bind to Synapses and Disrupt Synaptic Activity in a Manner That Requires APP [J]. The Journal of neuroscience : the official journal of the Society for Neuroscience, 2017, 37(49): 11947-66.